# HAMSTER: Hierarchical Action Models for Open-World Robot Manipulation

**Yi Li**[⋆‡1,2]**, Yuquan Deng**[⋆2]**, Jesse Zhang**[⋆1,3]**, Joel Jang**[1,2]**, Marius Memmel**[2]** Caelan Garrett**[1]**,**
**Fabio Ramos**[1]**, Dieter Fox**[1,2]**, Anqi Li**[†1]**, Abhishek Gupta**[†1,2]**, Ankit Goyal**[†1]
[1]NVIDIA  [2]University of Washington  [3]University of Southern California

## Abstract

Large foundation models have shown strong open-world generalization to complex problems in vision and language, but similar levels of generalization have yet to be achieved in robotics. One fundamental challenge is the lack of robotic data, which are typically obtained through expensive on-robot operation. A promising remedy is to leverage cheaper, "off-domain" data such as action-free videos, hand-drawn sketches or simulation data. In this work, we posit that *hierarchical* vision-language-action (VLA) models can be more effective in utilizing off-domain data than standard monolithic VLA models that directly finetune vision-language models (VLMs) to predict actions. In particular, we study a class of hierarchical VLA models, where the high-level VLM is finetuned to produce a coarse 2D path indicating the desired robot end-effector trajectory given an RGB image and a task description. The intermediate 2D path prediction is then served as guidance to the low-level, 3D-aware control policy capable of precise manipulation. Doing so alleviates the high-level VLM from fine-grained action prediction, while reducing the low-level policy's burden on complex task-level reasoning. We show that, with the hierarchical design, the high-level VLM can transfer across significant domain gaps between the off-domain finetuning data and real-robot testing scenarios, including differences on embodiments, dynamics, visual appearances and task semantics, etc. In the real-robot experiments, we observe an average of 20% improvement in success rate across seven different axes of generalization over OpenVLA, representing a 50% relative gain. Visual results are provided at: https://hamster-robot.github.io/

## 1 Introduction

Developing general robot manipulation policies has been notoriously difficult. With the advent of large vision-language models (VLMs) that display compelling generalization capabilities, there is optimism that the same recipe is directly applicable to robot manipulation. A line of prior work (Brohan et al., 2023a; Kim et al., 2024; Black et al., 2024) builds open-world vision-language-action models (VLAs) by finetuning off-the-shelf pretrained VLMs to directly produce robot actions. These VLA models, which we refer to in this work as *monolithic* VLA models, rely crucially on large robotics datasets, complete with on-robot observations, e.g., images and proprioceptive states, and actions. However, on-robot data is expensive, since end-to-end observation-action pairs are typically collected on the robot hardware through, e.g., teleoperation. Despite recent community-wide efforts in building large-scale robotics datasets (Collaboration et al., 2023; Khazatsky et al., 2024), the size, quality, and diversity of existing robotics datasets are still limited, and monolithic VLA models have yet to demonstrate emergent capability comparable to VLMs and LLMs in other domains of study. Moreover, monolithic VLA models are constrained by their inference frequency to achieve dexterous and dynamic manipulation tasks (Brohan et al., 2023a; Kim et al., 2024).

On the other hand, relatively small robot policy models have shown impressive dexterity and robustness. Such models have demonstrated promise across a range of complex tasks involving contact-rich manipulation and 3D reasoning, spanning domains from tabletop manipulation (Shridhar et al.,

---

[⋆] co-first authors [‡] project lead [†] equal advising

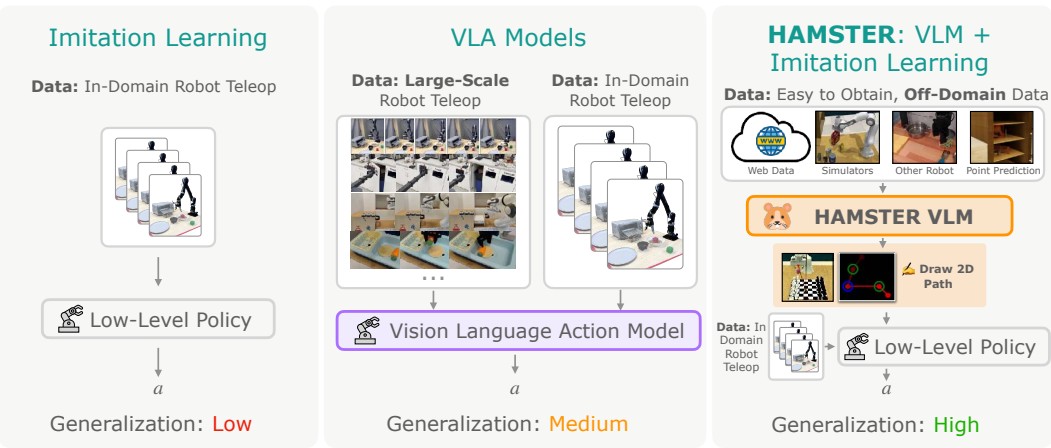

Figure 1: Overview of HAMSTER, VLAs and "smaller" imitation learning methods. HAMSTER's hierarchical design results in better generalization with a small amount of in-domain data. HAMSTER is able to utilize cheap training sources such as videos or simulations for enhanced generalization.

2023; Goyal et al., 2023; 2024; Ke et al., 2024) to fine dexterous manipulation (Chi et al., 2023; Zhao et al., 2023). Trained on relatively small datasets, these models show local robustness, and can achieve dexterous and high-precision control. However, they are often brittle to drastic changes in the environment or semantic description of the tasks (Pumacay et al., 2024). These models also can struggle to effectively leverage simulation data for real-world manipulation tasks due to sim-to-real gaps in visual appearances and system dynamics (Li et al., 2024; Mandlekar et al., 2021).

In this work, we ask – how can we marry the generalization benefits of large VLMs, with the efficiency, local robustness, and dexterity of small policy models? Our key insight is that, instead of directly predicting robot actions, VLMs can be fine-tuned to produce intermediate representations as high-level guidance on solving the robot manipulation task. The intermediate representation can then be consumed by the low-level policy model to produce actions, alleviating the low-level policy from the burden of long-horizon planning and complex, semantic reasoning. Further, if the intermediate representations are chosen such that they are *1)* easily obtainable from image sequences; *2)* largely embodiment agnostic; and *3)* sufficiently robust to subtle changes in dynamics, the VLM can be fine-tuned with *off-domain* data where robot actions are unavailable or inaccurate. Such off-domain data does not need to be collected on the actual robot hardware. Examples of off-domain data include action-free video data, simulation data, human videos, and videos of robot with different embodiments. These off-domain data are generally easier to collect and may already be abundant in existing datasets. We hypothesize, and show experimentally in Fig 7, that this hierarchical separation can allow VLA models to more effectively bridge the domain gap between off-domain data and in-domain robotic manipulation.

To this end, we propose a hierarchical architecture for VLAs, HAMSTER (**H**ierarchical **A**ction **M**odels with **S**epara**TE**d Path **R**epresentations), where large fine-tuned VLMs are connected to low-level policy models via 2D path representations[1]. A 2D path is a coarse trajectory of the 2D image-plane position of the robot end-effector[2], as well as where the gripper state changes, i.e., opens and closes (see Fig. 2). These 2D paths can be obtained cheaply and automatically from data sources such as action-free videos or physics simulations, using point tracking (Doersch et al., 2023; Karaev et al., 2025), hand-sketching (Gu et al., 2023), or proprioceptive projection. This allows HAMSTER can effectively leverage these abundant and inexpensive off-domain data when fine-tuning the high-level VLM. The hierarchical design presented in HAMSTER also offers additional advantages through the decoupling of VLM training and low-level action prediction. Specifically, while the higher-level VLM is predicting semantically meaningful trajectories from monocular RGB camera inputs, the lower-level policy models can additionally operate from rich 3D and proprioceptive inputs. In doing so, HAMSTER inherits the semantic reasoning benefits of VLMs along with the 3D

---

[1]Representations similar to 2D paths has been explored in the robot learning literature (Gu et al., 2023), primarily as a technique for flexible task specification. We refer readers to section 2 for a detailed discussion.

[2]For human video, this corresponds to the position of the palm center or fingertips.

reasoning and spatial awareness benefits of 3D policy models (Goyal et al., 2024; Ke et al., 2024). Moreover, the high-level VLM and low-level policy model can be queried at different frequencies

In summary, we study a family of hierarchical VLA models HAMSTERs, where finetuned VLMs are connected to low-level 3D policy models (Goyal et al., 2024; Ke et al., 2024). The 2D paths produced by high-level VLMs serve as guidance for a low-level policy that operates on rich 3D and proprioceptive inputs, allowing low-level policies to focus on robustly generating precise, spatially-aware actions. In our experiments, we observe an average of 20% improvement in success rate over seven different axes of generalization over OpenVLA (Kim et al., 2024), which amounts to 50% relative gain, as shown in Table 6. Since HAMSTER is built on both open-source VLMs and low-level policies, it can serve as a fully open-sourced enabler for the community-building vision-language-action models. It is important to note that while we are certainly not the first to propose hierarchical VLA models (Gu et al., 2023; Nasiriany et al., 2024a), we propose the novel insight that this type of hierarchical decomposition allows for these models to make use of abundant off-domain data for improving real-world control. This opens the door to alternative ways of training large vision-language-action models using cheaper and more abundant data sources.

## 2 RELATED WORK

**LLMs and VLMs for robotics.** Early attempts in leveraging LLMs and VLMs for robotics are through pretrained language (Jang et al., 2022; Shridhar et al., 2023; Singh et al., 2023) and visual (Shah & Kumar, 2021; Parisi et al., 2022; Nair et al., 2023; Ma et al., 2023) representations. However, these are insufficient for complex semantic reasoning and generalization to the open world (Brohan et al., 2022; Zitkovich et al., 2023). Recent research has focused on directly leveraging open world reasoning and generalization capability of LLMs and VLMs, by prompting or fine-tuning them to, e.g., generate plans (Duan et al., 2024; Huang et al., 2023b; Lin et al., 2023; Liang et al., 2023; Singh et al., 2023; Brohan et al., 2023b) or construct value (Huang et al., 2023a) and reward functions (Kwon et al., 2023; Sontakke et al., 2023; Yu et al., 2023; Ma et al., 2024; Wang et al., 2024). Our work is more closely related to VLA models, summarized below.

**Monolithic VLA models as language-conditioned robot policies.** Monolithic VLA models have been proposed to produce robot actions given task description and image observations directly (Brohan et al., 2022; Jiang et al., 2023; Zitkovich et al., 2023; Team et al., 2024; Kim et al., 2024; Radosavovic et al., 2023). Monolithic VLA models are often constructed from VLMs (Liu et al., 2024d; Bai et al., 2023; Driess et al., 2023; Lin et al., 2024), and are trained on large-scale on-robot data (Brohan et al., 2022; Collaboration et al., 2023; Khazatsky et al., 2024) to predict actions as text or special tokens. However, due to the lack of coverage in existing robotics datasets, they must be finetuned in-domain on expensive on-robot data. Their action frequency is also constrained by inference frequency, limiting their capability to achieve dexterous and dynamic tasks. The most relevant monolithic VLA model to our work is LLARVA (Niu et al., 2024), which predicts end-effector trajectories in addition to robot actions. However, LLARVA only uses trajectory prediction as an auxiliary task to improve the action prediction of a monolithic VLA model. In contrast, our work takes a hierarchical approach, enabling us to use specialist lower-level policies that take in additional inputs the VLMs cannot support, such as 3D pointclouds, to enable better imitation learning. Our predicted paths then enable these lower-level policies to generalize more effectively.

**VLMs for predicting intermediate representations.** Our work bears connections to prior methods using vision-language models to predict intermediate representations. These methods can be categorized by the choice of predicted representations:

*Point-based predictions:* A common intermediate prediction interface has been keypoint affordances (Stone et al., 2023; Sundaresan et al., 2023; Nasiriany et al., 2024b; Yuan et al., 2024b; Kuang et al., 2024). Keypoint affordances can be obtained through using open-vocabulary detectors (Minderer et al., 2022), iterative prompting of VLMs (Nasiriany et al., 2024b), or fine-tuning detectors to identify certain parts of an object by semantics (Sundaresan et al., 2023). Perhaps most related to our work, Yuan et al. (2024b) finetune a VLM to predict objects of interest as well as free space for placing an object, and Liu et al. (2024b) propose a mark-based visual prompting procedure to predict keypoint affordances as well as a fixed number of waypoints. As opposed to these, our work finetunes a VLM model to not just predict points but rather entire 2D paths, making it more broadly applicable across robotic tasks.

**Trajectory-based predictions:** The idea of using trajectory-based task specifications to condition low-level policies was proposed in RT-trajectory (Gu et al., 2023), largely from the perspective of flexible task specification. This work also briefly discusses the possibility of combining trajectory-conditioned model with trajectory sketches generated by a pre-trained VLM. Complementary to RT-Trajectory, the focus of this work is less on the use of trajectory sketches for task specification, but rather a hierarchical design of VLAs such that the high-level VLM can be fine-tuned with relative cheap and abundant data sources. This could include data such as action-free videos, or simulation data that look very different from the real world. We show that the emergent generalization capability of VLMs from its web-scale pretraining allows it transfer to test scenarios of interest with considerable visual and semantic variations. While RT-trajectory uses human effort or off-the-shelf pre-trained VLMs to generate trajectories, we show that fine-tuning VLM models on cheap data sources can generate significantly more accurate and generalizable trajectories (see Table. 5). Moreover, our instantiation of this architecture enables the incorporation of rich 3D and proprioceptive information, as compared to monocular 2D policies (Gu et al., 2023).

Similarly, the emergence of track-any-point (TAP) models (Doersch et al., 2023; Wang et al., 2023) has enabled policies conditioned on object trajectories (Yuan et al., 2024a; Xu et al., 2024; Bharadhwaj et al., 2024) or points sampled from a fixed grid in the image (Wen et al., 2023). While our current formulation focuses on end-effector trajectories, this framework can naturally extend to predicting object trajectories or other motion cues. By leveraging the predictive capabilities of VLMs, such an extension could further enhance the model's ability to generalize across diverse scenarios and improve its capacity for fine-grained motion reasoning.

**Leveraging simulation data for training robot policies.** There has been extensive work on leveraging simulation for robot learning. Simulation data is popular in reinforcement learning (RL), as RL on real robotic systems is often impractical due to high sample complexity and safety concerns (Lee et al., 2020; Handa et al., 2023; Torne et al., 2024). Recently, simulation has been also exploited to directly generate (Fishman et al., 2022) or bootstrap (Mandlekar et al., 2023) large-scale datasets for imitation learning, to reduce the amount of expensive robot teleoperation data needed. Our work takes a different approach – using simulation data to finetune a VLM, and showing that VLM is able to transfer the knowledge learned from simulation data to real robot systems, despite considerable visual differences. A related observation is recently made by (Yuan et al., 2024b), but they use keypoint affordances as the interface between the VLM and the low-level policy as opposed to more general expressive 2D path representations.

## 3 BACKGROUND

**Imitation Learning via Supervised Learning.** Imitation learning trains a policy $\pi_\theta(a \mid s, o, z)$ from expert demonstrations, where $s$ denotes proprioceptive inputs, $o$ includes perceptual observations (e.g., RGB images, depth), and $z$ provides task instructions. Given an expert dataset $\mathcal{D} = \{(s_i, o_i, z_i, a_i)\}_{i=1}^N$, the policy is optimized via maximum likelihood estimation, maximizing $\mathbb{E}_{(s_i, o_i, z_i, a_i) \sim \mathcal{D}} [\log \pi_\theta (a_i \mid s_i, o_i, z_i)]$. Despite advancements in architectures such as 3D policy representations (Goyal et al., 2023; Ke et al., 2024), generalizing to novel semantic or visual variations remains challenging. In this paper, we explore how VLMs can enhance imitation learning models for better generalization.

**Vision-Language Models.** VLMs (Liu et al., 2024a; Lin et al., 2024; Liu et al., 2024d) are large transformer models (Vaswani et al., 2023) that accept both vision and text tokens to generate text responses. They are pre-trained on extensive multimodal datasets (Zhu et al., 2023; Byeon et al., 2022) and later fine-tuned on high-quality, task-specific data (Shen et al., 2021; Lu et al., 2022). By tokenizing each modality into a shared space, these models autoregressively produce sequences of text tokens conditioned on an image and prior tokens. In our work, we assume access to such a pre-trained, text-and-image VLM (Lin et al., 2024; Liu et al., 2024d), further fine-tuned via a supervised loss that minimizes the negative log-likelihood of the target tokens.

## 4 HAMSTER: HIERARCHICAL ACTION MODELS FOR ROBOTIC LEARNING

In this work, we examine how VLA models can leverage relatively abundant data and demonstrate cross-domain transfer capabilities, as opposed to relying purely on expensive observation-language-

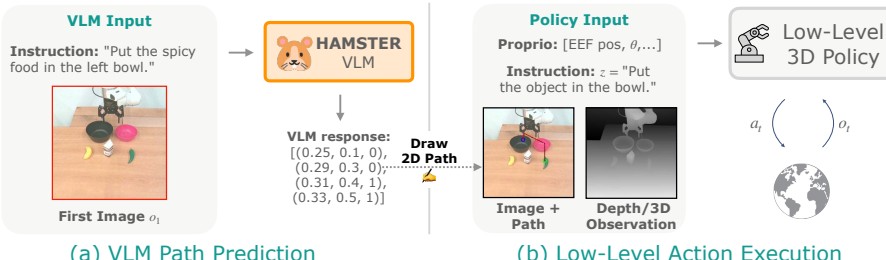

(a) VLM Path Prediction    (b) Low-Level Action Execution

Figure 2: Depiction of HAMSTER's execution. The high-level VLM is called once to generate the 2D path. The low-level policy is conditioned on the 2D path and interacts with the environment sequentially to execute low-level actions. The path predicted by the VLM enhances the low-level policy generalization capability.

action data collected on a robot. HAMSTER is a family of hierarchical VLA models designed for this purpose, exhibiting generalizable and robust manipulation. It consists of two interconnected models: first, a higher-level VLM that is finetuned on large-scale, off-domain data to produce intermediate 2D path guidance (detailed in Section 4.1), and second, a low-level policy that produces actions conditioned on 2D paths (detailed in Section 4.2).

The primary advantages of finetuning such a hierarchical VLM that produces intermediate representations as opposed to directly producing actions $a$ with a monolithic model (Kim et al., 2024; Zitkovich et al., 2023; Black et al., 2024) are threefold: *1)* our hierarchical VLM can leverage off-domain datasets lack of precise actions, e.g., simulation and videos; *2)* we find empirically that hierarchical VLMs producing 2D paths generalize more effectively cross-domain than monolithic VLA models; and *3)* the hierarchical design provides more flexibility on the sensory modality, and allows for asynchronous query of large high-level VLA models and small low-level policy models.

### 4.1   HAMSTER'S VLM FOR PRODUCING 2D PATHS TRAINED FROM OFF-DOMAIN DATA

The high-level VLM of HAMSTER predicts a coarse 2D path $p$ to achieve the task given a monocular RGB image `img` and language instruction $z$, i.e., $\hat{p} \sim \mathtt{VLM}(\mathtt{img}, z)$. The 2D path $p$ describes a coarse trajectory of the robot end-effector, or human hand in the case of human videos, on the input camera image. It also contains information about the gripper state. Formally, the 2D path is defined as $p = [(x_t, y_t, \mathtt{gripper\_open}_t)]_t$ where $x_t, y_t \in [0, 1]$ are *normalized pixel locations* of the end effector's (or hand) position at step $t$, and $\mathtt{gripper\_open}_t$ is a binary value indicating the gripper state, i.e., open and close.

Although, any pretrained text-and-image-input VLM (Lin et al., 2024; Liu et al., 2024d; Achiam et al., 2023) can be used to predict such a 2D path by casting an appropriate prompt, we find that pre-trained VLMs struggle with predicting such a path in a zero-shot manner (see Table 5). Therefore, we finetune pre-trained VLMs on datasets that ground VLMs to robot scenes and path predictions collected from easier-to-obtain sources, i.e., internet visual-question-answering data, robot data from other modalities, and simulation data. This is in contrast to work such as Gu et al. (2023), where pre-trained VLMs are tasked with directly performing spatially relevant path generation.

We use VILA-1.5-13b (Lin et al., 2024) as our base VLM, a 13-billion-parameter vision language model trained on interleaved image-text datasets and video captioning data. Although it is possible to curate a dataset on path prediction $\{(\mathtt{img}_i, z_i, p_i)\}_i$ and train the VLM *only* on the dataset, the literature (Brohan et al., 2023a; Yuan et al., 2024b) has shown that *co-training* the VLM on a variety of relevant tasks, all framed as VQA tasks, can help retain the VLM's generalization capability. To this end, we curate a multi-domain dataset to finetune this model for effective 2D path prediction.

#### 4.1.1   FINETUNING OBJECTIVE AND DATASETS.

Predicting the 2D path of the end-effector requires understanding *what* objects to manipulate in a given task in terms of their pixel positions, but also reasoning about *how* a robot should perform the task. To enable this understanding, we collate a diverse off-domain dataset $\mathcal{D}_{\mathrm{off}}$ from a wide range of modalities, including real-world data, visual question-answering data, and simulation data. Importantly, *none* of this off-domain data used to train the VLM comes from the deployment environment, thereby emphasizing generalizability.

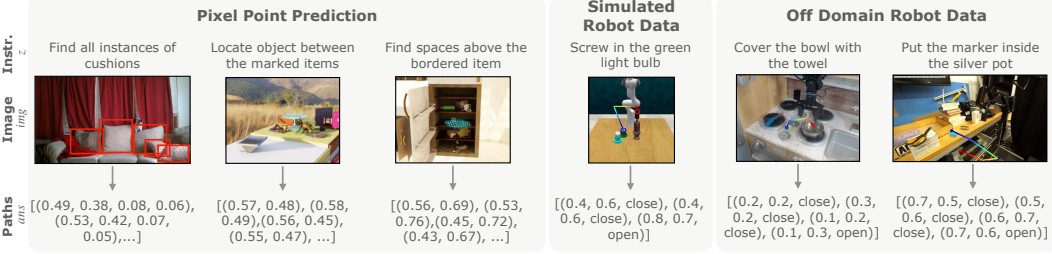

Figure 3: **Off Domain Training Data**: $\mathcal{D}_{\text{off}}$ contains (a) Pixel Point Prediction: 770k object location tasks from RoboPoint. (b) Simulated Robot Data: 320k 2D end-effector paths from RLBench environment. (c) Real Robot Data: 110k 2D end-effector paths from Bridge and DROID trajectories.

We assemble a dataset $\mathcal{D}_{\text{off}} = \{(\text{img}_i, z_i, \text{ans}_i)\}_{i=1}^{M}$ of image inputs $\text{img}_i$, language prompts $z_i$, and answer $\text{ans}_i$ consisting of three types of *off-domain* data: (1) pixel point prediction tasks (*what*); (2) simulated robotics tasks (*what and how*); (3) a real robot dataset consisting of trajectories (*what and how*). We detail each dataset below; see Figure 3 for visualization of each dataset's prompts and corresponding answers.

**Pixel Point Prediction.** For pixel point prediction, we use the RoboPoint dataset (Yuan et al., 2024b) with 770k pixel point prediction tasks, with most answers represented as a list of 2D points corresponding to locations on the image. A sample consists of a prompt $z$ like `Locate object between the marked items`, an input image $\text{img}$ and answer $\text{ans}$ like $[(0.25, 0.11), (0.22, 0.19), (0.53, 0.23)]$.[3] See the left of Figure 3 for an example. This dataset consists of data automatically generated in simulation and collected from existing real-world datasets; its diverse tasks enable the HAMSTER VLM to reason about pixel-object relationships across diverse scenes while retaining its semantic generalization capabilities.

**Simulated Robot Data.** We additionally generate a dataset of simulated robotics tasks from RL-Bench (James et al., 2020), a simulator of a Franka robot performing tabletop manipulation for a wide array of both prehensile and non-prehensile tasks. We use the simulator's built-in planning algorithms to automatically generate successful manipulation trajectories. Given a trajectory, we use the first frame from the front camera as the image input $\text{img}$. We construct prompt $z$ to instruct the VLM to provide a sequence of points denoting the trajectory of the robot gripper to achieve the given language instruction (see Figure 2). The ground-truth 2D path $p = [(x_t, y_t, \text{gripper\_open}_t)]_t$ is given by propriceptive projection using forward kinematics and camera parameters.

We generate 1000 episodes for each of 81 robot manipulation tasks in RLBench, each episode with $\sim 4$ language instructions, for a total of around $320k$ ($\text{img}, z, \text{ans}$) tuples, where $\text{ans} = p$. See the middle of Figure 3 for an example.

**Real Robot Data.** Using real robot data allows us to ensure the VLM can reason about objects and robot gripper paths when conditioned on scenes, including real robot arms. We use existing, online robot datasets *not from the deployment environment* to enable this VLM ability. We source 10k trajectories from the Bridge dataset (Walke et al., 2023; Collaboration et al., 2023) consisting of a WidowX arm (different embodiment from test robot) performing manipulation tasks and around 45k trajectories from DROID (Khazatsky et al., 2024). We covert both datasets to VQA dataset in as similar way as the simulated RL-Bench data, where the 2D paths are extracted from proprioception and camera parameters (see the right of Figure 3 for an example). Note that we essentially utilize the robot data as video data, where the end effector is tracked over time. In principle, this could be done with any number of point-tracking methods (Doersch et al., 2023) on raw video as well, with no action or proprioceptive labels.

We finetune the HAMSTER VLM on all three types of data by randomly sampling from all samples in the entire dataset with equal weight. We also include a 660k-sample VQA dataset (Liu et al., 2024c) for co-training to preserve world knowledge. We train with the standardized supervised prediction loss to maximize the log-likelihood of the answers $\text{ans}$:
$$\mathbb{E}_{(\text{img}_i, z_i, \text{ans}_i) \sim \mathcal{D}_{\text{off}}} \log \text{VLM}(\text{ans}_i \mid \text{img}_i, z_i).$$

---

[3]Note that this is not a temporally ordered path, but rather a set of unordered points of interest in an image.

**Remark.** One issue with simulation and real robot data is that the extracted 2D paths $p$ can be extremely long, e.g., exceeding one hundred steps. Since we want the HAMSTER VLM to reason at a *high level* instead of on the same scale as the low-level control policy, we simplify the paths $p^o$ with the Ramer-Douglas-Peucker algorithm (Ramer, 1972; Douglas & Peucker, 1973) that reduces curves composed of line segments to similar curves composed of fewer points. We refer readers to Appendix G for an ablation study.

## 4.2 PATH GUIDED LOW-LEVEL POLICY LEARNING

The low-level policy of HAMSTER $\pi_\theta(a \mid s, o, z, p)$ is conditioned on proprioceptive and perceptive observations, (optional) language instruction and, importantly, 2D path. While a low-level control policy *can* learn to solve the task without 2D path, the paths allow the low-level policy to forgo long-horizon and semantic reasoning and focus on local and geometric predictions to produce robot actions. As we find empirically (see Figure 4), 2D paths allow for considerably improved visual and semantic generalization of low-level policies.

HAMSTER's general path-conditioning framework allows lower-level policies to take in proprioceptive and perceptual (e.g., depth images) observations, that are not input to the high-level VLM. We consider low-level policies based on 3D perceptual information, i.e., $o = (\texttt{img}, \texttt{pointcloud})$, available at test time on a robotic platform with standard depth cameras. We study two choices of policy architecture, RVT-2 (Goyal et al., 2024) and 3D-DA (Ke et al., 2024) which has shown state-of-the-art results on popular robot manipulation benchmark (James et al., 2020).

**Conditioning on Paths.** Most policy architectures use the form $\pi_\theta(a \mid s, o, z)$ without 2D path inputs. One naïve option is to concatenate the path with proprioceptive or language inputs. However, because 2D paths vary in length, the architecture must handle variable-length inputs. To incorporate the 2D path $\hat{p}$ from the VLM without major modifications, we alternatively overlay the 2D path onto the image observation (Gu et al., 2023). Our implementation follows this approach by drawing colored trajectories on all images in the trajectory $o_i^1, \ldots, o_i^T$: points at each $(x_t, y_t)$ are connected with line segments using a color gradient to indicate temporal progression (see Figure 2(b)), and circles mark changes in gripper status (e.g., green for closing, blue for opening). If the policy architecture allows images with more than three channels, we can also include path drawing as separate channels, instead of overlaying it on the RGB channel. We empirically study both drawing strategies, overlay and concatenating channels, in section 5.3.

**Policy Training.** To train the policy, we collect a relatively small-scale task-specific dataset $\mathcal{D} = \{(s_i, o_i, z_i, a_i)\}_{i=1}^N$ on the robot hardware. During training, we use *oracle* 2D paths constructed by proprioception projection, similar to how the 2D paths are constructed for the VLM training data, and construct path-labeled dataset $\mathcal{D}_{\text{path}} = \{(s_i, o_i, z_i, p_i, a_i)\}_{i=1}^N$. We train a policy $\pi_\theta(a \mid s, o, z, p)$ with standard supervised imitation learning objectives on $\mathcal{D}_{\text{path}}$ to maximize the log-likelihood of the dataset actions: $\mathbb{E}_{(s_i, o_i, z_i, p_i, a_i) \sim \mathcal{D}_{\text{path}}} \log \pi_\theta(a_i \mid s_i, o_i, z_i, p_i)$. For further implementation details, see Appendix B.

**Inference Speed.** Monolithic VLAs query the VLM at every action step (Kim et al., 2024; Brohan et al., 2023a), which can be very expensive with large VLMs. For example, OpenVLA's 7B-parameter VLA only runs at 6Hz on an RTX 4090 (Kim et al., 2024). Instead, HAMSTER's hierarchical design allows us to query the VLM only one or few times during an episode to generate 2D paths $\hat{p}$ that can be followed by low-level policy for multiple steps. Therefore, HAMSTER can be scaled to large VLM backbones without needing end-users to be concerned about inference speed.

## 5 EXPERIMENTAL EVALUATION

We evaluate our approach in both simulation and real-world experiments to the following key questions. Do hierarchical VLAs:

**Q1** Generalize behaviors to unseen scenarios with significant visual and semantic variation?
**Q2** Achieve stronger cross-domain generalization than monolithic architectures?
**Q3** Facilitate learning of non-prehensile and long-horizon tasks?
**Q4** Exhibit strong demonstration efficiency?
**Q5** Have improved visual + semantic reasoning due to hierarchy and VLM fine-tuning?

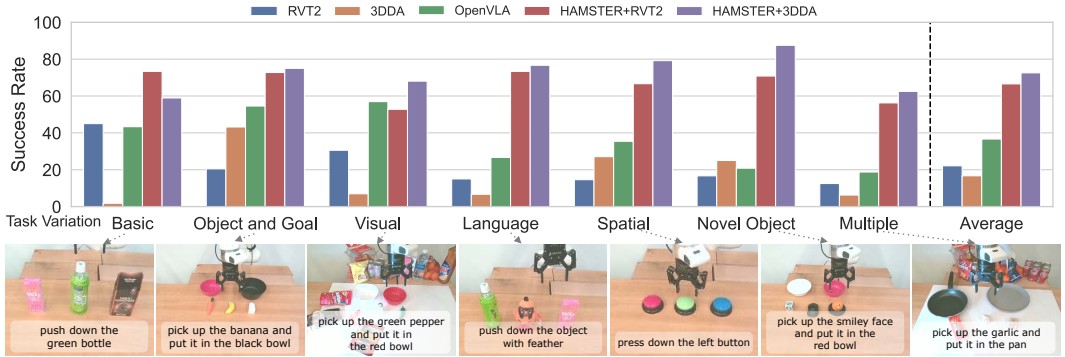

Figure 4: Depiction of quantitative real-world policy execution results on a real-world robot, evaluated across different axes of generalization and across both prehensile and non-prehensile tasks. Across all generalization axes, HAMSTER outperforms monolithic VLAs and the base 3D imitation learning policies.

## 5.1 REAL WORLD EVALUATION ON TABLETOP MANIPULATION

To answer **Q1**, our real-world evaluation experiments aim to test the generalization capability of hierarchical VLA models across significant semantic and visual variations. In particular, we consider a variant of HAMSTER that uses a VLM (VILA-1.5-13b (Lin et al., 2024)) finetuned on the data mixture in Section 4.1 as the high-level predictor, with two low-level 3D policy architectures - RVT-2 (Goyal et al., 2024) and 3D Diffuser Actor (3D-DA) (Ke et al., 2024) as choices of the low-level policy, as described in Section 4.2. The low-level 3D policies are trained with 320 episodes collected via teleoperation shown in Fig. 3. Importantly, the high-level VLM has not seen any in-domain data and is only finetuned on the off-domain data described in Section 4.1. This suggests that any generalization that the VLM shows result from cross-domain transfer.

**Baseline comparisons.** To answer **Q2**, we compare HAMSTER with a state-of-the-art monolithic VLA, OpenVLA (Kim et al., 2024) as well as non-VLM 3D policies, RVT-2 (Goyal et al., 2024) and 3D-DA (Ke et al., 2024). For fair comparison, we finetune OpenVLA on the collected in-domain data described above since OpenVLA showed poor zero-shot generalization. The 3D policy (RVT-2, 3D-DA) baselines are trained with the same teleoperation data used to train the low-level policy in HAMSTER but without the intermediate 2D path representation from HAMSTER's VLM.

**Finetuning OpenVLA with RLBench.** To ensure our method's advantage over OpenVLA (Kim et al., 2024) is not solely due to RLBench data, we fine-tuned OpenVLA on the same RLBench dataset used for HAMSTER's VLM—1,000 episodes per task across 81 tasks (using only episodes with good front-camera visibility)—until achieving over 90% token accuracy (Kim et al., 2024). We then fine-tuned this model on our tasks following the procedure in Appendix C.2. In real-world pick-and-place experiments (6 trials over 6 "Basic" tasks as shown in Table 4), RLBench-finetuned OpenVLA averaged a success score of 0.54 versus 0.58 for the model without RLBench fine-tuning. This suggests that monolithic VLA architectures like OpenVLA gain little benefit from RLBench data, likely due to mismatches in action and observation spaces relative to the real-world setup.

**Quantitative Results.** Figure 4 summarizes our real-world results. To answer **Q3**, we evaluate across multiple task types, including 'pick and place,' and nonprehensile tasks such as 'press buttons' and 'knock down objects.' We also test generalization across various axes (**Q1**) – *obj and goal:* unseen object-goal combinations; *visual:* visual changes in table texture, lighting, distractor objects; *language:* unseen language instructions (e.g., candy → sweet object); *spatial:* unseen spatial object relationships in the instruction; *novel object:* unseen objects; and lastly, *multiple:* a combination of multiple variations. In total, we evaluate each model on 74 tasks for 222 total evaluations. Detailed results and the success score metric are provided in Appendix Table 4.

**Qualitative Eval on Various Tasks.** In addition to the quantitative evaluation conducted for comparison with OpenVLA, we also present qualitative results that demonstrate how HAMSTER's hierarchical structure enables low-level policy models to generalize to more complex tasks. Figure 8 illustrates the diverse tasks HAMSTER can handle, including unfolding a towel, opening and closing drawers, pressing buttons, wiping surfaces, and cleaning tables. These tasks present challenges such as varying lighting conditions, cluttered backgrounds, and semantic understanding requiring exter-

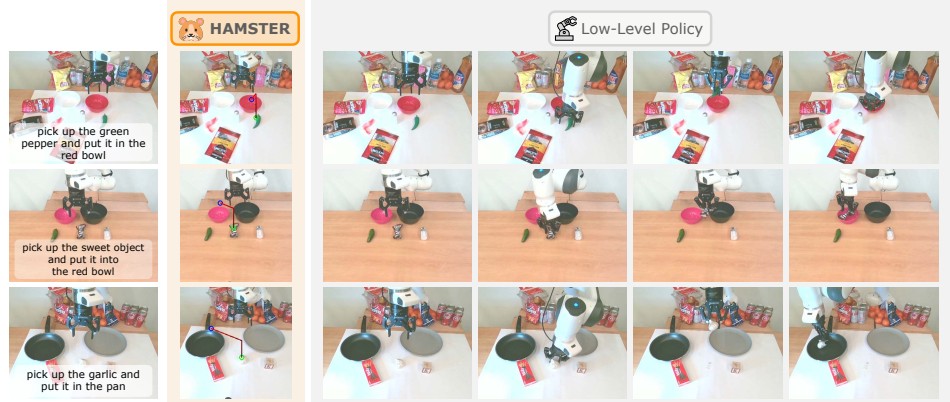

Figure 5: Example real-world HAMSTER rollouts demonstrate its strong performance in novel scenes achieved by leveraging VLMs' generalization capabilities and the robust execution of low-level 3D policies.

| Method | Success |
|---|---|
| 3D-DA | $0.18 \pm 0.10$ |
| HAMSTER+3D-DA (50%) | $0.36 \pm 0.04$ |
| HAMSTER+3D-DA | $\mathbf{0.43 \pm 0.05}$ |

Table 1: Results on Colosseum demonstrate that HAMSTER is data efficient, achieving 2X the success score of 3D-DA with just 50% of the data.

| Method | Original Camera | | Novel Camera | |
|---|---|---|---|---|
| | Success | Complete | Success | Complete |
| OpenVLA | 0.60 | 0.30 | 0.23 | 0.00 |
| HAMSTER+RVT2 | 0.83 | 0.70 | 0.73 | 0.40 |
| HAMSTER+RVT2 (Concat) | **1.00** | **1.00** | **0.98** | **0.90** |

Table 2: Real world results demonstrate HAMSTER generalizes to better to novel camera views (see Fig.Figure 6). We ran 10 trails and report averaged success score (success) described in Table 4 and number of successful executions (complete).

nal world knowledge. Additionally, HAMSTER demonstrates the ability to perform long-horizon tasks—none of which are part of the in-domain training set used to train the policy model.

Overall, we find that HAMSTER significantly outperforms monolithic VLA models and (non-VLM) 3D policies by over **2x** and **3x**, respectively, on average. This is significant because this improved performance is in the face of considerable visual and semantic changes in the test setting, showing the ability of HAMSTER to generalize better than monolithic VLA models or non-VLM base models. We further group results by task type in Table 6, where we see HAMSTER outperforms OpenVLA across all task types (pick and place, press button, and knock down). See Appendix C for evaluation conditions, a task list, and other experiment details, and Appendix E for failure modes.

## 5.2 SIMULATION EVALUATION

**Overall Results.** For further investigation into **Q1**, **Q2**, and **Q3**, we conducted a controlled simulation evaluation using Colosseum (Pumacay et al., 2024), which provides significant visual and semantic variations across pick-place and non-prehensile tasks. Pairing our high-level VLM with the state-of-the-art 3D-DA (Ke et al., 2024) policy on RLBench, we compared HAMSTER against a vanilla 3D-DA implementation without path guidance. As shown in Table 3 over 5 seeds, HAMSTER outperforms the vanilla approach by an average of 31%. This improvement stems from training with path-drawn images, which encourages the policy to focus on the path rather than extraneous visual features, thereby enhancing robustness to visual variations. We refer readers to Pumacay et al. (2024) for details on the variations and Appendix F for further simulation experiment details.

**HAMSTER with Fewer Demonstrations.** We also test HAMSTER's ability to work well with limited demonstrations to answer **Q4**. We test on a subset of 5 Colosseum tasks, namely, SLIDE_BLOCK_TO_TARGET, PLACE_WINE_AT_RACK_LOCATION, INSERT_ONTO_SQUARE_PEG, STACK_CUPS, SETUP_CHESS. Results in Table 1 demonstrate that HAMSTER+3D-DA with just 50% of the data still achieves 2x the success rate of standard 3D-DA, demonstrating that HAMSTER is demonstration-efficient for the downstream imitation learning tasks.

## 5.3 VLM GENERALIZATION STUDIES

Finally, we answer **Q5**: can HAMSTER's hierarchy enable superior visual and semantic reasoning?

**Camera View Invariance.** We test HAMSTER+RVT2 against OpenVLA from a new camera angle (Figure 6) across 10 pick-and-place trials using 6 training objects and 3 training containers to

|  | Avg. | no var | bac tex | cam pos | distractor | lig col | man obj col | man obj siz |
|---|---|---|---|---|---|---|---|---|
| 3D-DA[ Ke et al.] | 0.35 ± 0.04 | 0.43 ± 0.06 | 0.34 ± 0.07 | 0.35 ± 0.11 | 0.39 ± 0.11 | 0.44 ± 0.13 | 0.41 ± 0.04 | 0.41 ± 0.11 |
| HAMSTER (w 3D-DA) | **0.46** ± 0.04 | **0.57** ± 0.03 | **0.48** ± 0.08 | **0.39** ± 0.06 | **0.41** ± 0.05 | **0.59** ± 0.04 | **0.57** ± 0.08 | **0.51** ± 0.10 |
|  | man obj tex | rec obj col | rec obj siz | rec obj tex | rlb and col | rlb var | tab col | tab tex |
| 3D-DA[ Ke et al.] | 0.27 ± 0.04 | 0.34 ± 0.10 | 0.36 ± 0.05 | 0.36 ± 0.12 | 0.07 ± 0.03 | 0.45 ± 0.12 | 0.42 ± 0.06 | 0.23 ± 0.04 |
| HAMSTER (w 3D-DA) | **0.48** ± 0.06 | **0.48** ± 0.05 | **0.40** ± 0.05 | **0.56** ± 0.09 | **0.11** ± 0.10 | **0.58** ± 0.04 | **0.56** ± 0.03 | **0.35** ± 0.07 |

Table 3: Simulation evaluation of HAMSTER across different visual variations. We test vanilla 3D Diffuser Actor and HAMSTER across variations in Colosseum (Pumacay et al., 2024) and find that HAMSTER generalizes more effectively than 3D Diffuser Actor. Avg. indicates mean across variations, including no variation.

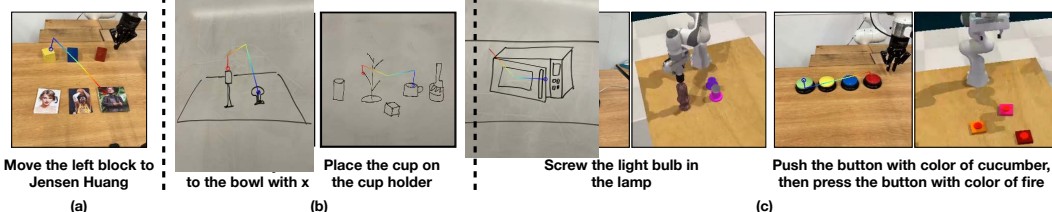

**Move the left block to Jensen Huang**
(a)

**to the bowl with x**

**Place the cup on the cup holder**
(b)

**Screw the light bulb in the lamp**

**Push the button with color of cucumber, then press the button with color of fire**
(c)

Figure 7: HAMSTER's VLM demonstrates strong generalization to unseen scenarios. From left to right: (a) leveraging world knowledge for user-specified tasks, (b) handling out-of-domain inputs like human-drawn sketches, and (c) transferring from diverse simulations to visually distinct real-world tasks. Blue-to-red lines indicate motion, with blue and red circles marking grasp and release points, respectively.

check HAMSTER's visual spatial reasoning. The results in Table 2 show that HAMSTER significantly outperforms OpenVLA and remains robust to new camera angles, benefiting from its VLM trained on diverse *off-domain* tasks across various viewpoints. Additionally, we compare HAMSTER+RVT2 (Concat), where instead of overlaying the path on the input RGB image, we modify RVT-2 to accept a 6-channel input by concatenating the original RGB image with a separate RGB image containing only the drawn path. We can easily apply this due to HAMSTER's hierarchical nature. Concatenated paths actually achieve the best performance, demonstrating the effectiveness of this path representation, though it is less general and not compatible with all imitation learning policy architectures (such as 3D-DA as it uses a pre-trained image encoder expecting 3 input channels). One possible explanation is that RVT2's virtual reprojection can fragment the 2D path when it is directly drawn on the image, making it harder for RVT2 to decode. By providing a dedicated path channel (via concatenation), path guidance is preserved more effectively.

**VLM Generalization.** We further demonstrate the benefit of HAMSTER's hierarchy by demonstrating that the VLM generalizes well to visually unique and semantically challenging tasks due to its off-domain fine-tuning. We visualize example HAMSTER path drawings in Figure 7, demonstrating HAMSTER's VLM itself effectively reasons semantically and visually for unseen tasks. We further investigate VLM performance in Appendix D.1, where we find that (1) HAMSTER outperforms zero-shot path generation from closed-source VLMs (Gu et al., 2023; Liang et al., 2023) and (2) that inclusion of simulation data improves HAMSTER's real-world performance. Both results point to the benefit of explicit hierarchy: off-domain VLM fine-tuning that improves its performance. See Appendix D.1 for further details.

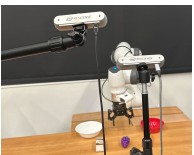

Figure 6: Camera pos. for view invariance: old (right) and new (left).

## 6 CONCLUSION AND LIMITATIONS

In summary, we study hierarchical VLA models that achieve robust generalization in robotic manipulation. We introduce HAMSTER, consisting of a finetuned VLM that accurately predicts 2D paths and a low-level policy that learns to generate actions using the 2D paths. This two-step architecture enables visual generalization and semantic reasoning across considerable domain shifts while enabling specialist policies, like ones conditioned on 3D inputs, to execute low-level actions.

This work represents an initial step towards developing versatile, hierarchical VLA methods. The proposed work only generates points in 2D space, without making native 3D predictions. This prevents the VLM from having true spatial 3D understanding. Moreover, the interface of just using 2D paths is a bandwidth limited one, which cannot communicate nuances such as force or rotation. In the future, investigating learnable intermediate interfaces is a promising direction. Moreover, training these VLMs directly from large-scale human video datasets would also be promising.

ACKNOWLEDGEMENTS

We thank Wentao Yuan for generously providing the Robopoint dataset. We also acknowledge Entong Su and Yunchu Zhang for their assistance in setting up the robot environment. We are grateful for the support from the Army Research Lab through sponsored research, as well as the Amazon Science Hub for Yi and Marius. We also thank Animesh Garg for many helpful discussions. Finally, we extend our gratitude to Yao Lu, Hongxu Yin, Ligeng Zhu, Borys Tymchenko, and Zhijian Liu from NVIDIA's VILA group for their valuable support throughout this work.

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

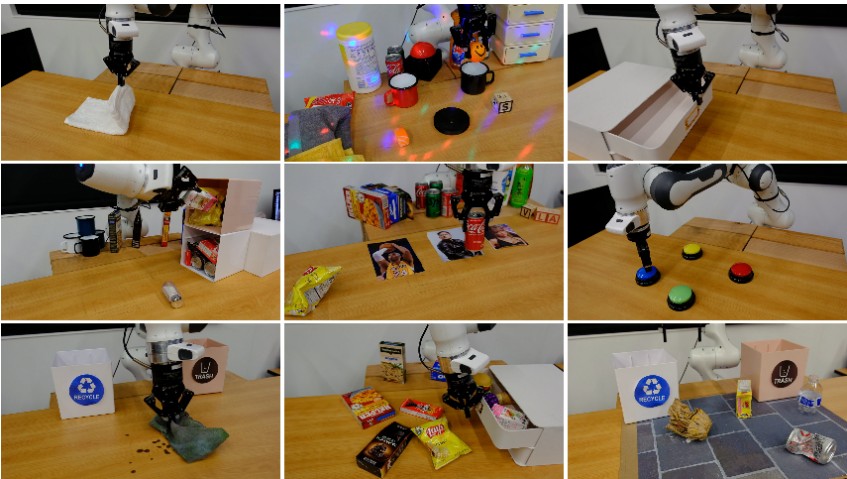

Figure 8: Examples of various robot tasks and environments that HAMSTER can handle. See more details in our teaser video at https://hamster-robot.github.io/.

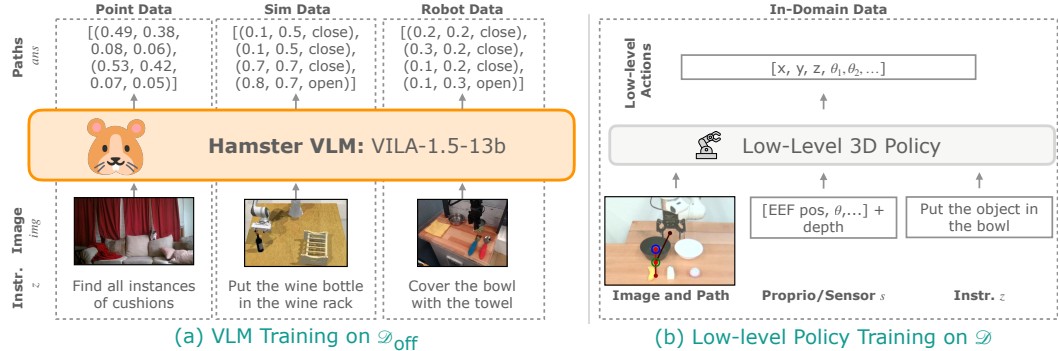

Figure 9: (a): Examples of training data in $\mathcal{D}_{\text{off}}$ used to train HAMSTER's VLM. (b): The data used to train HAMSTER's low-level policies.

## A  VLM FINETUNING DATASET DETAILS

**Pixel Point Pred Data.**   Our point prediction dataset comes from Robopoint (Yuan et al., 2024b). 770k samples in our point prediction dataset contain labels given as a set of unordered points such as $p^o = [(0.25, 0.11), (0.22, 0.19), (0.53, 0.23)]$, or bounding boxes in $[(cx, cy, w, h)]$ style. Other than that, following Robopoint (Yuan et al., 2024b), we use the VQA dataset Liu et al. (2024c) with 660k samples which answer VQA queries in natural language such as "What is the person feeding the cat?" We keep these data as is because these VQA queries are likely to benefit a VLM's semantic reasoning and visual generalization capabilities; we fine-tune HAMSTER's VLM on the entire Robopoint dataset as given.

**Simulation Data.**   We selected 81 RLBench tasks out of 103 to generate data by removing tasks with poor visibility on the `front_cam` view in RLBench. We use the first image in each episode combined with each language instruction. The final dataset contains around 320k trajectories.

**Real Robot Data.**   For the Bridge (Walke et al., 2023) dataset, which only provides RGB images, we extract trajectories by iteratively estimating the extrinsic matrix for each episode. In each scene, we randomly sample a few frames and manually label the center of the gripper fingers. Using the corresponding end-effector poses, we compute the 3D-2D projection matrix with a PnP (Perspective-n-Point) approach. We then apply this projection matrix to the episodes and manually check for any

misalignments between the projected gripper and the actual gripper. Episodes exhibiting significant deviations are filtered out, and a new round is started to estimate their extrinsic matrix.

For DROID (Khazatsky et al., 2024), a large portion of the dataset contains noisy camera extrinsics information that do not result in good depth alignment. Therefore, we filter out trajectories with poor-quality extrinsics as measured by the alignment between the projected depth images and the RGB images. This results in ∼45k trajectories (∼22k unique trajectories as trajectories each have 2 different camera viewpoints) which we use for constructing the VLM dataset $\mathcal{D}_{\text{off}}$ as described in Section 4.1.

## B   IMPLEMENTATION AND ARCHITECTURE DETAILS

---

**HAMSTER Prompt**

In the image, please execute the command described in ⟨quest⟩{quest}⟨/quest⟩.
Provide a sequence of points denoting the trajectory of a robot gripper to achieve the goal.
Format your answer as a list of tuples enclosed by ⟨ans⟩ and ⟨/ans⟩ tags. For example:

```
⟨ans⟩[(0.25, 0.32), (0.32, 0.17), (0.13, 0.24), ⟨action⟩Open
Gripper⟨/action⟩, (0.74, 0.21), ⟨action⟩Close Gripper⟨/action⟩,
...]⟨/ans⟩
```

The tuple denotes the $x$ and $y$ location of the end effector of the gripper in the image. The action tags indicate the gripper action.
The coordinates should be floats ranging between 0 and 1, indicating the relative locations of the points in the image.

---

Figure 10: The full text prompt we use to train HAMSTER with on simulation and real robot data (Section 4.1). We also use this prompt for inference.

### B.1   VLM IMPLEMENTATION DETAILS

**VLM Prompt.**   We list the prompt for both fine-tuning on sim and real robot data and evaluation in Figure 10. We condition the model on an image and the prompt, except when training on Pixel Point Prediction data (i.e., from Robopoint (Yuan et al., 2024b)) where we used the given prompts from the dataset. Note that we ask the model to output gripper changes as separate language tokens, i.e., Open Gripper/Close Gripper, as opposed to as a numerical value as shown in simplified depictions like Figure 2.

**VLM Trajectory Processing.**   As mentioned in Section 4.1, one problem with directly training on the path labels $p^o$ is that many paths may be extremely long. Therefore, we simplify the paths $p^o$ with the Ramer-Douglas-Peucker algorithm (Ramer, 1972; Douglas & Peucker, 1973) that reduces curves composed of line segments to similar curves composed of fewer points. We run this algorithm on paths produced by simulation and real robot data to generate the labels $p^o$ for $\mathcal{D}_{\text{off}}$. We use tolerance $\epsilon = 0.05$, resulting in paths that are around 2-5 points for each short horizon task.

**VLM Training Details.**   We train our VLM, VILA1.5-13B Lin et al. (2024), on a node equipped with eight NVIDIA A100 GPUs, each utilizing approximately 65 GB of memory. The training process takes about 30 hours to complete. We use an effective batch size of 256 and a learning rate of $1 \times 10^{-5}$. During fine-tuning, the entire model—including the vision encoder—is updated.

### B.2   LOW-LEVEL POLICY TRAINING DETAILS

We train RVT2 (Goyal et al., 2024) and 3D-DA (Ke et al., 2024) as our lower-level policies. We keep overall architecture and training hyperparameters the same as paper settings. Specific details about how the inputs were modified other than the 2D path projection follow.

For low-level policy training, we train the policies on ground truth paths constructed by projecting trajectory end-effector points to the camera image. In order to also ensure the policies are robust to possible error introduced by HAMSTER VLM predictions during evaluation, we add a small

---

**RT-Trajectory GPT-4o Prompt**

In the image, please execute the command described in '{quest}'.

Provide a sequence of keypoints denoting a trajectory of a robot gripper to achieve the goal. Keep in mind these are keypoints, so you do not need to provide too many points.

Format your answer as a list of tuples enclosed by `<ans>` and `</ans>` tags. For example:

`<ans>[(0.25, 0.32), (0.32, 0.17), (0.13, 0.24), <action>Open Gripper</action>, (0.74, 0.21), <action>Close Gripper</action>, ...]</ans>`

The tuple denotes point $x$ and $y$ location of the end effector of the gripper in the image. The action tags indicate the gripper action.

The coordinates should be floats ranging between 0 and 1, indicating the relative locations of the points in the image.

The current position of the robot gripper is: {current_position}. Do not include this point in your answer.

---

Figure 11: The full text prompt we use to prompt RT-Trajectory with GPT4-o.

amount of random noise ($N(0, 0.01)$) to the 2D path $(x, y)$ image points during training to obtain slightly noisy path drawings. No noise was added to the gripper opening/closing indicator values.

**RVT2 (Goyal et al., 2024).** We remove the language instruction for RVT-2 when conditioning on HAMSTER 2D paths.

**3D-DA (Ke et al., 2024).** In simulated experiments in Colosseum, no changes were needed. In fact, we saw a performance drop for HAMSTER+3D-DA when removing language for Colosseum tasks and a small drop in performance when using simplified language instructions. This is likely due to 3D-DA's visual attention mechanism which cross attends CLIP language token embeddings with CLIP visual features, therefore detailed language instructions are beneficial.

In real-world experiments, we simplify the language instruction in the same way as for RVT2 when conditioning on HAMSTER 2D paths to encourage following the trajectory more closely with limited data. In addition, we reduced the embedding dimension of the transformer to 60 from 120, removed proprioception information from past timesteps, and reduced the number of transformer heads to 6 from 12 in order to prevent overfitting.

## C  REAL WORLD EXPERIMENT DETAILS

### C.1  TRAINING TASKS AND DATA COLLECTION

For our real-world experiments, we collected all data using a Franka Panda arm through human teleoperation, following the setup described in Khazatsky et al. (2024). Below, we describe the training tasks:

**Pick and place.** We collected 220 episodes using 10 toy objects. In most of the training data, 2 bowls were placed closer to the robot base, while 3 objects were positioned nearer to the camera. The language goal for training consistently followed the format: `pick up the {object} and put it in the {container}`.

**Knock down objects.** We collected 50 episodes with various objects of different sizes. Typically, 3 objects were arranged in a row, and one was knocked down. The language goal for training followed the format: `push down the {object}`.

**Press button.** We collected 50 episodes with 4 colored buttons. In each episode, the gripper was teleoperated to press one of the buttons. The language goal followed the format: `press the {color} button`.

---

**RT-Trajectory Code as Policies Prompt**

Task Instruction: {task_instruction}
Robot Constraints:

- The robot arm takes as input 2D poses with gripper open/closing status of the form $(x, y, \text{gripper\_open} == 1)$
- The gripper can open and close with only binary values
- The workspace is a $1 \times 1$ square centered at $(0.5, 0.5)$
- The x-axis points rightward and y-axis points downward.

Please write Python code that generates a list of 2D poses and gripper statuses for the robot to follow. Include Python comments explaining each step. Assume you can use `numpy` or standard Python libraries, just make sure to import them.

Enclose the start and end of the code block with `` and `` so that it can be parsed. Make sure that it is a self-contained script such that when executing the code string, there is a variable named `robot_poses` which is a list of poses of the form: `[(x, y, gripper), (x, y, gripper), ...]`.

Scene Description:

```

{scene_description}

```

Figure 12: The full text prompt we use for RT-Trajectory with Code-as-Policies on top of GPT4-o. The scene description at the bottom comes from an open-vocabulary object detector describing each detected object and its bounding box in the image based on the task instruction.

When training RVT2, which requires keyframes as labels, in addition to labeling frames where the gripper performs the `open gripper` and `close gripper` actions, we also included frames that capture the intermediate motion as the gripper moves toward these keyframes.

## C.2 BASELINE TRAINING DETAILS

**OpenVLA (Kim et al., 2024).** Following Kim et al. (2024), we only utilize parameter efficient fine-tuning (LoRA) for all of our experiments, since they showed that it matches full fine-tuning performance while being much more efficient. We follow the recommended default rank of $r$=32. We opt for the resolution of 360 x 360 to match all of the baseline model's resolutions. We also follow the recommended practice of training the model until it surpasses 95% token accuracy. However, for some fine-tuning datasets, token accuracy converged near 90%. We selected the model checkpoints when we observed that the token accuracy converged, which usually required 3,000 to 10,000 steps using a global batch size of either 16 or 32. Training was conducted with 1 or 2 A6000 gpus (which determined the global batch size of 16 or 32). Emprically, we observed that checkpoints that have converged showed very similar performance in the real world. For example, when we evaluate checkpoint that was trained for 3,000 steps and showed convergence, evaluating on a checkpoint trained for 5,000 steps of the same run resulted in a very similar performance.

**RT-Trajectory (Gu et al., 2023).** We implement two versions of RT-Trajectory for the comparison in Table 5. The first (0-shot GPT-4o) directly uses GPT-4o to generate 2D paths with a prompt very similar to the one we use for HAMSTER, displayed in Figure 11.

The second version implements RT-Trajectory on top of a Code-as-Policies (Liang et al., 2023), as described in RT-Trajectory. We use OWLv2 (Minderer et al., 2023) to perform open-vocabulary object detection on the image to generate a list of objects as the scene description and then prompt RT-Trajectory with the prompt shown in Figure 12. We also use GPT-4o as the backbone for this method.

| Category | Task | OpenVLA | RVT2 | RVT2+Sketch | 3DDA | 3DDA+Sketch |
|---|---|---|---|---|---|---|
| Basic | pick up the corn and put it in the black bowl | 1 | 1 | 1 | 0 | 0.25 |
| Basic | pick up the grape and put it in the white bowl | 1 | 0.75 | 1 | 0 | 1 |
| Basic | pick up the milk and put it in the white bowl | 0 | 1 | 1 | 0 | 0.25 |
| Basic | pick up the salt bottle and put it in the white bowl | 0.75 | 0.5 | 1 | 0 | 0 |
| Basic | pick up the shrimp and put it in the red bowl | 0.75 | 0.5 | 1 | 0 | 1 |
| Basic | pick up the cupcake and put it in the red bowl | 0 | 0.5 | 0.5 | 0.25 | 1 |
| Basic | press down the red button | 0.5 | 0 | 1 | 0 | 1 |
| Basic | press down the green button | 0 | 1 | 0 | 0 | 0.25 |
| Basic | press down the yellow button | 0 | 0 | 1 | 0 | 1 |
| Basic | press down the blue button | 0.5 | 0 | 1 | 0 | 0.5 |
| Basic | push down the green bottle | 0.5 | 0 | 0.5 | 0 | 1 |
| Basic | push down the pocky | 0 | 1 | 1 | 0 | 0.5 |
| Basic | push down the red bag | 0.5 | 0.5 | 0 | 0 | 0.5 |
| Basic | push down the bird toy | 0 | 0 | 0 | 0 | 0.5 |
| Basic | push down the yellow box | 1 | 0 | 1 | 0 | 0.5 |
| Object and Goal | pick up the salt bottle and put it in the white bowl | 1 | 1 | 1 | 0.5 | 1 |
| Object and Goal | pick up the banana and put it in the black bowl | 0.25 | 0.25 | 1 | 0.5 | 1 |
| Object and Goal | pick up the grape and put it in the black bowl | 1 | 0.25 | 0.5 | 1 | 1 |
| Object and Goal | pick up the carrot and put it in the red bowl | 0.75 | 0 | 1 | 0.5 | 1 |
| Object and Goal | pick up the milk and put it in the white bowl | 0.25 | 0 | 1 | 0 | 0.25 |
| Object and Goal | pick up the shrimp and put it in the white bowl | 0.25 | 0.75 | 0.5 | 0.25 | 1 |
| Object and Goal | pick up the cupcake and put it in the black bowl | 0.25 | 0 | 1 | 0.5 | 0.75 |
| Object and Goal | pick up the icecream and put it in the black bowl | 0.25 | 0 | 0.5 | 0.5 | 1 |
| Object and Goal | pick up the corn and put it in the red bowl | 1 | 0 | 1 | 1 | 1 |
| Object and Goal | pick up the green pepper and put it in the red bowl | 0.75 | 0 | 0.5 | 0 | 0.25 |
| Object and Goal | pick up the orange and put it in the white bowl | 0.25 | 0 | 0 | 0 | 0 |
| Visual(Table Texture) | pick up the salt bottle and put it in the white bowl | 1 | 1 | 1 | 0 | 1 |
| Visual(Table Texture) | pick up the banana and put it in the black bowl | 0.25 | 0.25 | 0.75 | 0.5 | 0.75 |
| Visual(lighting) | pick up the grape and put it in the black bowl | 0.25 | 0 | 0.5 | 0.25 | 0 |
| Visual(lighting) | pick up the carrot and put it in the red bowl | 0.75 | 0 | 1 | 0 | 0.75 |
| VIsual(clutter) | pick up the milk and put it in the white bowl | 0.75 | 0.25 | 1 | 0.25 | 1 |
| VIsual(clutter) | pick up the shrimp and put it in the red bowl | 0.75 | 0.5 | 0 | 0 | 0.5 |
| Visual(mix) | pick up the green pepper and put it in the red bowl | 0.25 | 0 | 1 | 0 | 0.25 |
| Visual(mix) | pick up the salt bottle and put it in the white bowl | 0.25 | 0 | 0.25 | 0.25 | 1 |
| Visual(appearance change) | pick up the green pepper and put it in the red bowl | 1 | 0 | 0.5 | 0 | 1 |
| Visual(appearance change) | pick up the salt bottle and put it in the black bowl | 1 | 1 | 1 | 0 | 1 |
| Visual(Table Texture) | press down the red button | 1 | 1 | 0 | 0 | 0.5 |
| Visual(lighting) | press down the green button | 1 | 0 | 0.5 | 0 | 0.5 |
| VIsual(clutter) | press down the yellow button | 0 | 0 | 0.5 | 0 | 0.5 |
| Visual(mix) | press down the blue button | 0 | 0 | 0 | 0 | 0.5 |
| Visual(Table Texture) | push down the pocky | 0 | 1 | 0 | 0 | 0 |
| VIsual(clutter) | push down the green bottle | 1 | 0.5 | 1 | 0 | 1 |
| VIsual(clutter) | push down the chocolate box | 1 | 0 | 0 | 0 | 1 |
| Visual(mix) | push down the green bottle | 0 | 0 | 0.5 | 0 | 1 |
| Language | pick up the sweet object and put it in the red bowl | 1 | 1 | 1 | 0 | 1 |
| Language | pick up the spicy object and put it in the red bowl | 1 | 0 | 1 | 0 | 0.75 |
| Language | pick up the salty object and put it in the red bowl | 0 | 0 | 1 | 0 | 1 |
| Language | pick up the object with color of cucumber and put it in the red bowl | 0 | 0 | 1 | 0.25 | 0.75 |
| Language | pick up the object with color of lavender and put it in the black bowl | 0 | 0 | 1 | 0 | 1 |
| Language | pick up the object with the color of sky and and put it in the container with the color of coal | 1 | 0 | 0 | 0.25 | 1 |
| Language | pick up the block with the color of sunflower and put it in the container with the color of enthusiasm | 0 | 0.25 | 1 | 0 | 1 |
| Language | press the button with the color of fire | 0.5 | 0 | 1 | 0 | 0.5 |
| Language | press the button with the color of cucumber | 0 | 0 | 1 | 0 | 0.5 |
| Language | press the button with the color of sky | 0 | 0 | 0 | 0.5 | 1 |
| Language | press the button with the color of banana | 0 | 0 | 0 | 0 | 0.5 |
| Language | push down the object with color of leaf | 0 | 1 | 1 | 0 | 0 |
| Language | push down the box contains cruchy biscuit | 0 | 0 | 0 | 0 | 1 |
| Language | push down the bag with color of fire | 0 | 0 | 1 | 0 | 0.5 |
| Language | push down the object with feather | 0.5 | 0 | 1 | 0 | 1 |
| Spatial | pick up the left object and put it in the left bowl | 0 | 1 | 1 | 0.25 | 1 |
| Spatial | pick up the middle object and put it in the left bowl | 0 | 0 | 1 | 0 | 1 |
| Spatial | pick up the right object and put it in the left bowl | 1 | 0 | 0.5 | 0.25 | 0.5 |
| Spatial | pick up the left object and put it in the right bowl | 0.25 | 0.25 | 1 | 0.25 | 1 |
| Spatial | pick up the middle object and put it in the right bowl | 0 | 0 | 1 | 0 | 1 |
| Spatial | pick up the right object and put it in the right bowl | 0.5 | 0 | 1 | 0 | 1 |
| Spatial | press down the left button | 0.5 | 0 | 0 | 0 | 0.5 |
| Spatial | press down the middle button | 0 | 0 | 1 | 1 | 0.5 |
| Spatial | press down the right button | 0 | 0 | 1 | 1 | 1 |
| Spatial | push down the left object | 0.5 | 0 | 0 | 0 | 0 |
| Spatial | push down the middle object | 1 | 0.5 | 0 | 0 | 1 |
| Spatial | push down the right object | 0.5 | 0 | 0.5 | 0.5 | 1 |
| Novel Object | pick up the "R" and put it in the red bowl | 0 | 0 | 1 | 0 | 1 |
| Novel Object | pick up the boxed juice and put it in the red bowl | 0 | 0.75 | 0.75 | 1 | 1 |
| Novel Object | pick up the cholate bar and put it in the white bowl | 0.25 | 0 | 0.5 | 0.5 | 1 |
| Novel Object | pick up the smile face and put it in the red bowl | 1 | 0 | 1 | 0 | 1 |
| Novel Object | pick up the mouse and put it in the red bowl | 0 | 0.25 | 1 | 0 | 1 |
| Novel Object | pick up the 5 and put it in the white bowl | 0 | 0 | 0 | 0 | 0.25 |
| Multiple | pick up the lays chip and put it in the pan | 0.25 | 0.25 | 0.75 | 0 | 1 |
| Multiple | pick up the garlic and put it in then pan | 0.25 | 0 | 1 | 0 | 0.25 |
| Multiple | pick up the "K" and put it in the pan | 0.25 | 0 | 0.5 | 0 | 1 |
| Multiple | pick up the pocky and put it in the pan | 0 | 0.25 | 0 | 0.25 | 0.25 |

Table 4: Detailed results of real-world evaluation. The first column indicates the variation category, while the second column presents the language instruction. For the `pick and place` task, 0.25 points are awarded for each successful action: reaching the object, picking it up, moving it to the target container, and placing it inside. For the `knock down` task, 0.5 points are awarded for touching the correct object and successfully knocking it down. For the `press button` task, 0.5 points are awarded for positioning the gripper above the correct button and successfully pressing it.

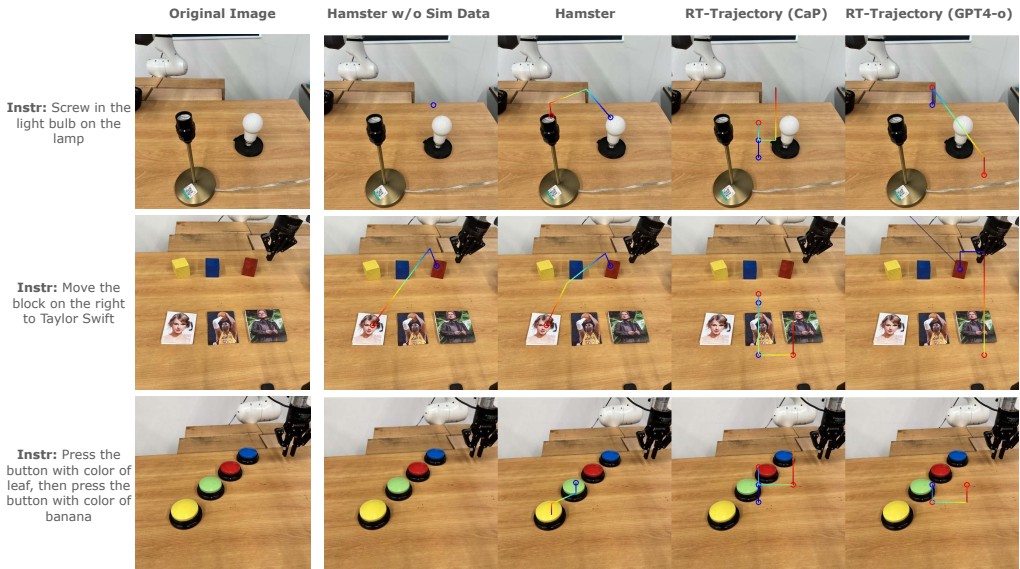

Figure 13: Human VLM evaluation example images and instructions along with corresponding trajectories from HAMSTER without any finetuning on (RLBench) simulation data, HAMSTER finetuned on all the data in Section 4.1, RT-Trajectory (Gu et al., 2023) with Code-as-Policies (Liang et al., 2023) powered by GPT-4o (Achiam et al., 2023), and RT-Trjaectory powered by GPT-4o directly.

## C.3 EVALUATION TASKS

We evaluate our method on the tasks of `pick and place`, `knock down object`, and `press button` across various generalization challenges, as illustrated in Figure 4. Detailed results are available in Table 4. Following (Kim et al., 2024), we assign points for each successful sub-action. For VLM, human experts are employed to assess the correctness of the predicted trajectories.

## D EXTENDED RESULTS

### D.1 IMPACT OF DESIGN DECISIONS ON VLM PERFORMANCE

To better understand the transfer and generalization performance of the proposed hierarchical VLA model, we analyze the impact of various decisions involved in training the high-level VLM. We conduct a human evaluation of different variants of a trained high-level VLM on a randomly collected dataset of real-world test images, as shown in Figure 7. We ask each model to generate 2D path traces corresponding to instructions such as "move the block on the right to Taylor Swift" or "screw the light bulb in the lamp" (the full set is in Appendix D.2). We then provide the paths generated by each method to human evaluators who have not previously seen any of the models' predictions. The human evaluators then rank the predictions for each method; we report the average rank across the samples in Table 5.

We evaluate the following VLM models: (1) zero-shot state-of-the-art closed-source models such as GPT-4o using a similar prompt to ours (shown in Figure 11), (2) zero-shot state-of-the-art closed-source models such as GPT-4o but using Code-as-Policies (Liang et al., 2023) to generate paths as described in Gu et al. (2023) (prompt in Figure 12), (3) finetuned open-source models (VILA-1.5-13b) on the data sources described in Section 4.1, but excluding the simulation trajectories from the RLBench dataset, (4) finetuned open-source models (VILA-1.5-13b) on the data sources described in Section 4.1, including path sketches from the RLBench dataset. The purpose of these evaluations is to first compare with closely related work that generates 2D trajectories using pretrained closed source VLMs Gu et al. (2023) (Comparison (1) and (2)). The comparison between (3) and (4) (our complete method) is meant to isolate the impact of including the simulation path sketches from the

| Method | VLM | Finetuning Data | Rank Exc. Real RLB. | Rank Real RLB. | Rank All |
|---|---|---|---|---|---|
| RT-Traj. | 0-shot GPT-4o | - | 3.40 | 3.63 | 3.47 |
| RT-Traj. | CaP GPT-4o | - | 3.57 | 3.36 | 3.41 |
| HAMSTER | VILA | Our Exc. Sim RLB. | 1.78 | 2.39 | 2.13 |
| HAMSTER | VILA | Our | **1.59** | **1.28** | **1.40** |

Table 5: Ranking-based human evaluation of different VLMs, averaged across various real-world evaluation tasks. Results indicate that HAMSTER including simulation data is most effective since it captures both spatial and semantic information across diverse tasks from RLBench. This significantly outperforms zero-shot VLM-based trajectory generation, as described in Gu et al. (2023)

RLBench dataset. In doing so, we analyze the ability of the VLM to predict intermediate paths to transfer across significantly varying domains (from RLBench to the real world).

The results suggest that: (1) zero-shot path generation, even from closed-source VLMs Gu et al. (2023) such as GPT-4o with additional help through Code-as-Policies (Liang et al., 2023), under-performs VLMs finetuned on cross-domain data as in HAMSTER; (2) inclusion of significantly different training data such as low-fidelity simulation during finetuning improves the real-world performance of the VLM. This highlights the transferability displayed by HAMSTER across widely varying domains. These results emphasize that the hierarchical VLA approach described in HAMSTER can effectively utilize diverse sources of cheap prior data for 2D path predictions, despite considerable perceptual differences.

## D.2 VLM REAL WORLD GENERALIZATION STUDY

The full list of task descriptions for this study is below (see Appendix D.1 for the main experiment details). Duplicates indicate different images for the same task. We plot some additional comparison examples in Figure 13. Note that the path drawing convention in images for this experiment differ from what is given to the lower-level policies as described in Section 4.2 as this multi-colored line is easier for human evaluators to see.

1. screw in the light bulb on the lamp
2. screw in the light bulb on the lamp
3. screw in the light bulb on the lamp
4. screw out the light bulb and place it on the holder
5. screw out the light bulb and place it on the holder
6. screw in the light bulb
7. screw in the light bulb on the lamp
8. move the blue block on Taylor Swift
9. pick up the left block and put it on Jensen Huang
10. move the block on the right to Taylor Swift
11. place the yellow block on Kobe
12. pick up the blue block and place it on Jensen Huang
13. move the red block to Kobe
14. press the button on the wall
15. press the button to open the left door
16. press the button to open the right door
17. open the middle drawer
18. open the bottom drawer
19. open the top drawer

20. open the middle drawer
21. open the bottom drawer
22. press the button
23. press the button
24. press the orange button
25. press the orange button with black base
26. press the button
27. pick up the SPAM and put it into the drawer
28. pick up the orange juice and put it behind the red box
29. pick up the tomato soup and put it into the drawer
30. pick up the peach and put it into the drawer
31. move the mayo to the drawer
32. move the dessert to the drawer
33. pick up the object on the left and place it on the left
34. pick up the fruit on the left and put it on the plate
35. pick up the milk and put it on the plate
36. press the button with the color of cucumber, then press the button with color of fire
37. press the button with color of banana
38. press the button with color of leaf
39. press the button with color of leaf, then press the one with color of banana
40. press left button
41. pick up the left block on the bottom and stack it on the middle block on top
42. make I on top of C
43. put number 2 over number 5
44. stack block with lion over block with earth
45. pick up the left block on the bottom and stack it on the middle block on top
46. stack the leftest block on the rightest block
47. stack the block 25 over block L
48. put the left block on first stair

## D.3 HUMAN RANKING

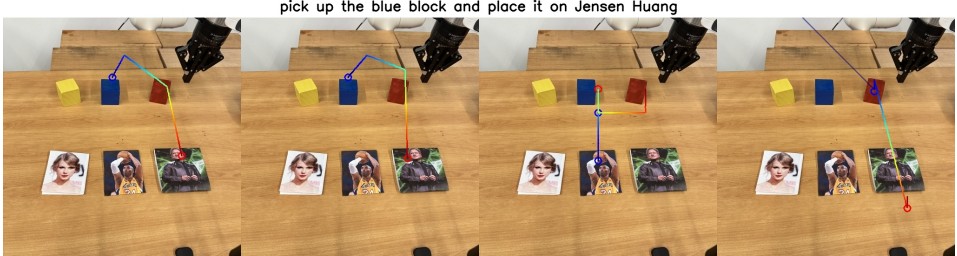

Figure 14: An example of results for human ranking. The trajectory is from blue to red with blue circle and red circle denotes gripper close point and open point respectively. The grader is asked to provide a rank to these trajectory about which trajectory has highest chance to succeed.

Due to the variety of possible trajectories that accomplish the same task, we use human rankings to compare how likely produced trajectories are to solve the task instead of quantitative metrics

such as MSE. To do that, we generate trajectories for 48 image-question pairs with HAMSTER w/o RLBench, HAMSTER, Code-as-Policy (Liang et al., 2023), and GPT4o (Achiam et al., 2023). See Figure 14 for an example.

We recruit 5 human evaluators, who are robot learning researchers that have not seen the path outputs of HAMSTER, to grade these 4 VLMs based on the instruction: *"Provide a rank for each method (1 for best and 4 for worst). In your opinion, which robot trajectory is most likely to succeed. Traj goes from blue to red, blue circle means close gripper, red circle means open gripper."* The evaluators are allowed to give multiple trajectories the same score if they believe those trajectories are tied. As they are robot learning researchers, they are familiar with the types of trajectories that are more likely to succeed. Therefore, these rankings act as a meaningful trajectory quality metric.

# E  FAILURE ANALYSIS

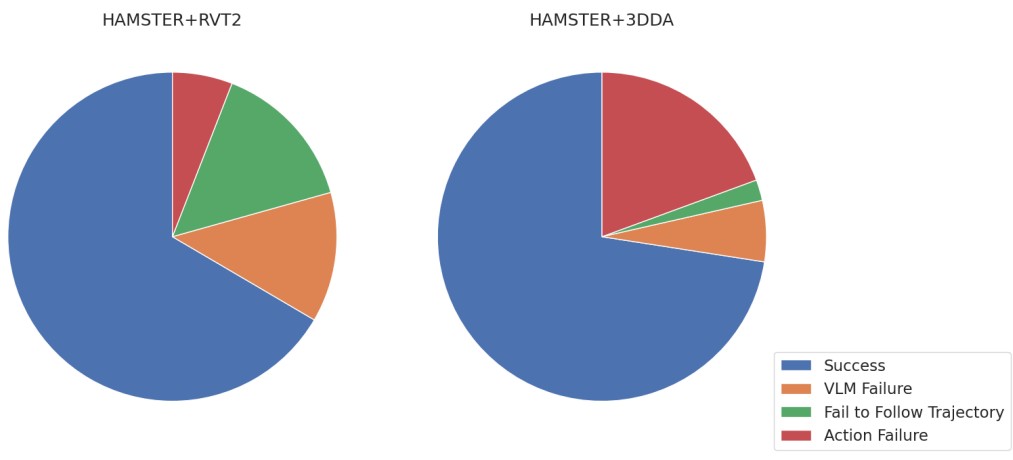

Figure 15: Performance Distribution of RVT2+Sketch and 3DDA+Sketch

This section outlines the failure modes observed during our experiments and provides a detailed breakdown of the causes. Failures can be attributed to issues in **trajectory prediction**, **trajectory adherence**, and **action execution**.

## E.1  DIFFERENT FAILURE MODES

**Trajectory Prediction Failures**   The Vision-Language Model (VLM) may fail to predict the correct trajectory due to several factors:

- *Failure to understand the language goal:* Although the VLM demonstrates strong capabilities in handling diverse task descriptions, it struggles when the training set lacks similar tasks. This can cause the model to misunderstand the goal and make inaccurate predictions.

- *Incorrect trajectory prediction:* In some cases, the VLM predicts an incorrect trajectory, either by interacting with the wrong objects or misinterpreting the direction of the affordance.

- *Dynamic changes in the environment:* Since trajectories are generated at the beginning of a task, significant environmental changes during execution can lead to failure. The model lacks the ability to dynamically adjust the trajectory or reidentify the object initially referenced.

**Trajectory Adherence Failures**   Failures in adhering to the predicted trajectory arise primarily due to:

- *3D ambiguity:* The use of 2D trajectory predictions introduces ambiguities, such as determining whether a point is positioned above or behind an object, leading to execution errors.

- *Incorrect object interaction:* The low-level action model is not explicitly constrained to strictly follow the predicted trajectory. As a result, it may deviate, interacting with the wrong object and causing task failures.

**Action Execution Failures**   Even when the trajectory is correctly predicted and adhered to, action execution may still fail due to:

- *Execution-specific issues:* Despite training on a diverse set of actions, the model may fail during execution. For example, in grasping tasks, an incorrect grasp angle can cause the object to slip, resulting in a failed grasp.

### E.2   FAILURE ANALYSIS

Our analysis in Figure 15 reveals distinct failure tendencies across methods.

For RVT, 72% of failures stemmed from the low-level model failing to follow the trajectory, while 28% were due to execution failures. In contrast, for 3DDA, only 10% of failures were related to trajectory adherence, with 90% attributed to execution failures.

We hypothesize that this discrepancy arises because RVT incorporates a re-projection step, complicating trajectory adherence. In contrast, 3DDA leverages a vision tower that processes the original 2D image, simplifying trajectory interpretation.

## F   SIMULATION EXPERIMENT DETAILS

Our simulation experiments are performed on Colosseum (Pumacay et al., 2024), a simulator built upon RL-Bench (James et al., 2020) containing a large number of visual and task variations to test the generalization performance of robot manipulation policies (see Figure 16 for a visualization of a subset of the variations). We use the `front_camera` and remove all tasks in which the camera does not provide a clear view of the objects in the task, resulting in 14 out of 20 colosseum tasks (we remove `basketball_in_hoop`, `empty_drawer`, `get_ice_from_fridge`, `move_hanger`, `open_drawer`, `turn_oven_on`).

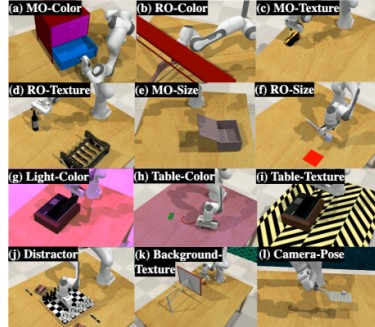

Figure 16: Colosseum benchmark variations. Figure from Pumacay et al. (2024), taken with permission.

Colosseum contains 100 training episodes for each task, without any visual variations, and evaluates on 25 evaluation episodes for each variation. We follow the same procedure other than using just the `front_camera` instead of multiple cameras. We report results in Table 3 after removing variations with no visual variations (e.g., object friction).

| Task | RVT2 | 3DDA | OpenVLA | HAMSTER+RVT2 | HAMSTER+3DDA |
|------|------|------|---------|--------------|--------------|
| pick and place | 0.28 | 0.19 | 0.46 | 0.79 | 0.78 |
| press button | 0.13 | 0.16 | 0.25 | 0.50 | 0.63 |
| knock down | 0.17 | 0.03 | 0.41 | 0.47 | 0.66 |

Table 6: Real world average success rates grouped by task type.

## G   DIFFERENT WAYS OF REPRESENTING 2D PATHS

To investigate the effect of the number of points on the 2D path, we train the VLM to predict 1. paths simplified using RDP algorithm, which simplify paths in short horizon tasks to 3-5 points and

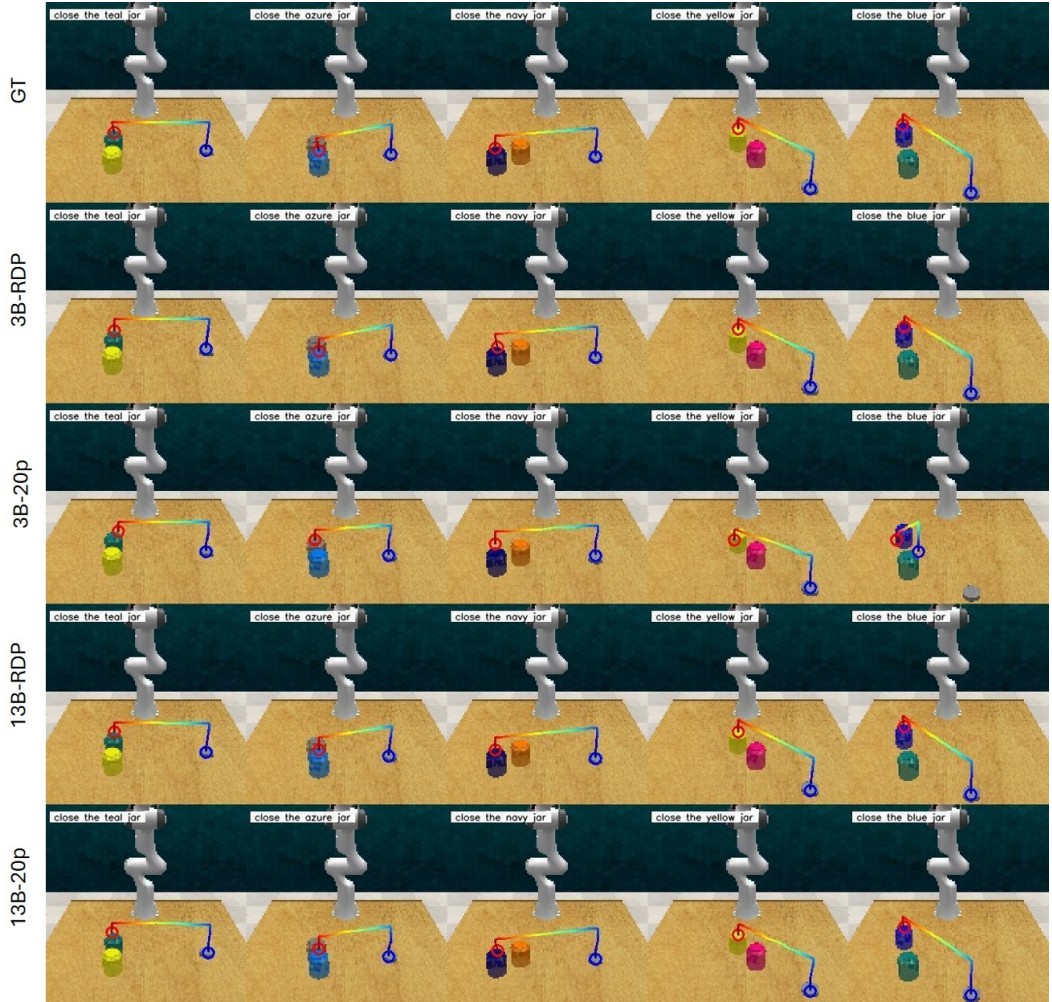

Figure 17: The task is to pick up the lid and close it on the jar with correct color. Task description is located on the top-left corner of each image. The trajectory goes from blue to red where blue circles denotes where the gripper should close and red circles denotes where the gripper should open. GT denotes ground truth, 3B and 13B denotes VILA1.5-3B and VILA1.5-13B, RDP denotes paths simplified using Ramer–Douglas–Peucker algorithm while 20p denotes paths reprensented using 20 points.

is what we used in the paper. We denote these paths as RDP in the following; 2. Paths represented with 20 points sampled on the path with same step size, denoted as 20p in the following. We keep points where the gripper is executing operation of open or close in both methods.

We train the network on RLBench 80 tasks with 1000 episodes for each task and test it on 25 episodes on the task of close jar. We tried both VILA1.5-3B (denoted as 3B) and VILA1.5-13B (denoted as 13B) as our backbone. Thus we have in total 4 combinations over 2 backbones and 2 designs of path representations. We visualize the result in this Figure 17.

From this result we can see that when using smaller models, like VILA1.5-3B, paths represented using points extracted using RDP algorithm outperforms paths represented with a fixed number of 20 points significantly. When the network becomes larger to the level of 13B, the VLM is able to handle the representation using 20 points and both two path representations work perfectly. We believe that is because when points are simplified using the RDP algorithm, we usually need less points to represent the path and helps the model to pay more attention to predict the accurate position for the gripper open/close points.

