# OpenReview forum: "HAMSTER: Hierarchical Action Models for Open-World Robot Manipulation"
_ICLR.cc/2025/Conference — ICLR 2025 Poster_

### Official Review · Reviewer_jRfz · 2024-11-01

**Soundness:** 2
**Presentation:** 3
**Contribution:** 2
**Rating:** 5
**Confidence:** 3

**Summary:**

This paper proposes a two-level hierarchical approach to solving robotic manipulation tasks.  Given an image and a language prompt, a vision-language model (VLM) predicts a 2D gripper trajectory in image space.  Then, a low-level policy generates end-effector motions conditioned on the 2D gripper trajectory.  The authors show that the VLM can be trained using large-scale, offline datasets without expensive, on-robot data.  The low-level policy is trained via imitation learning with small on-robot datasets.  The proposed method, called HAMSTER, is empirically evaluated on a range of real-world and simulated robot manipulation tasks.  HAMSTER outperforms baselines when generalizing to different language prompts, spatial rearrangements, or objects.  The 2D gripper trajectory predictions of HAMSTER's VLM is evaluated based on human evaluations of their accuracy, and it is shown to produce better predictions than the baselines.

**Strengths:**

- The paper demonstrates that a VLM can be trained on offline, cheap datasets to produce 2D gripper trajectories to solve a range of robotic manipulation problems not seen during training.  A human evaluation study shows that the proposed training scheme produces trajectory predictions that are better ranked than baselines.
- The proposed method, HAMSTER, combines this pre-trained VLM with a low-level policy to solve language-conditioned, long horizon manipulation tasks.  This hierarchical approach is shown to significantly outperform non-hierarchical baselines on real-world tasks, especially on variations not seen during training.
- HAMSTER leverages a large VLM in a way that is still efficient at inference time.  The VLM is queried once at the start, and then only the lighter-weight, low-level policy is used.

**Weaknesses:**

- The authors do not discuss common failure modes of the proposed approach.  It is written, "We refer readers to
the supplementary website for additional details on the failure modes and evaluation conditions."  This discussion is important for understanding the limitations of the approach and should be included in the main paper.  For instance, how often does the VLM produce infeasible paths? Does the low-level policy ever mess up a grasp but still follow the 2D trajectory?
- The results in the simulation evaluation are not compelling.  On average, HAMSTER outperforms 3D-DA by only 8%, even though HAMSTER's VLM was trained on RLBench (what Colleseum was built upon).   The discussion of this experiment is very brief so it is difficult to gain any insights.  For instance, why does HAMSTER have lower performance and much higher variance on the "no variation" task?  Is there a reason why one would expect the VLM to be especially helpful on tasks like "man obj size" or "tab col"?  Why did the authors switch from RVT-2 to 3D-DA for this setting?
- It is not clear why human ranking was used as an evaluation metric for the VLM's 2D trajectories.  No information is provided in the Appendix about how the participants were instructed to rank the trajectories, whether these participants were familiar with robot arms, nor how many participants were included.  Is there a reason that the 2D trajectories could not be evaluated based on mean squared error on a set of expert robot trajectories?

**Questions:**

- It's not clear how 'time' is encoded in the 2D trajectory image.  It is written "we use a color gradient to indicate progression through time."  Is this based on an absolute or relative scale?  Would two trajectories of different lengths end at the same color?  Can this approach encode the speed that the gripper should be moving?
- How was it decided that the VLM should generate 2D trajectories in image space **and** gripper apertures?  Given that the low-level policy has to predict 6DoF end effector motions, it does not seem unreasonable that it could also predict gripper motion? Alternatively, why not train the VLM to predict the rotation of the gripper (this information would be available in the Robot Simulation Data and the Real Robot Data)?
- Why do you only use the `front_cam` view in RLBench, as stated in Appendix A?  Given that the objective is to train a VLM that is robust to changes in view, it seems like the model would benefit from training on all camera views in RLBench.
- Most of the images of the experiments in the main paper show the a similar camera view (camera pointing toward robot base).  Did you evaluate the model on different camera views?  Do you have evidence or ideas on how HAMSTER would perform on different views?
- Why do you think 3D-DA performed worse when the language prompts were removed?

Minor typo: Missing period at bottom of page 4 "is similar An illustrative"

---

> ### Author Response · Authors · 2024-11-20
>
> Thank you for the detailed review! We are happy that you agree our approach significantly outperforms the baselines in the real world, and that we leverage a VLM in a computationally efficient manner. We respond to individual points below:
>
> **Q: Why do you only use the front_cam view in RLBench? Can HAMSTER generalize to multi-view?**
>
> A:  We only use front_cam as it is this view that contains best visibility of most tasks. Other standard camera angles like left_shoulder or right_shoulder usually have a large portion of the image blocked by the robot arm.
>
> Despite this, HAMSTER is quite robust to different views. We have the camera pointing toward the robot base because this view point has best visibility and is easy to set up, but the HAMSTER is robust to common viewpoints. See the below video for a demonstration:
> [video](https://drive.google.com/file/d/1nCbc2n2wLjgN0bKN9Rwe7RzCE0uFgGbY/view?usp=sharing)
>
> **Q: Why did the authors switch from RVT-2 to 3D-DA for this setting [simulated experiments]?**
>
> A: We used 3D-DA as it performed similarly to RVT-2 on RLBench, and at the time of writing, we encountered implementation issues with RVT-2 at the Colosseum. For consistency, we have now added results for HAMSTER with 3D-DA and the vanilla 3D-DA on the real robot in the  paper (see the updated Fig 3). Hence, we have now evaluated HAMSTER with 3D-DA both in sim and real. A brief results are shown below.
> | Task           | openvla | HAMSTER+RVT2 | HAMSTER+3DDA |
> |----------------|---------|--------------|--------------|
> | pick and place | 0.46    | **0.79**         | 0.78         |
> | press button   | 0.25    | 0.50         | **0.63**         |
> | knock down     | 0.41    | 0.47         | **0.66**         |
>
> **Q:  The authors do not discuss common failure modes of the proposed approach.**
>
> A:  Thank you for the suggestion. We have uploaded failure videos to the supplementary website and updated the failure analysis in Appendix E. System failures are categorized as follows:
>
> - **VLM Path Generation Failures** – Whether the VLM produces a reasonable path. [Video](https://drive.google.com/file/d/1omkkK9qaUtYYvx4Lb88InvGu5oI5tMFv/view?usp=drive_link)
> - **Path Following Failures** – Whether the low-level policy follows the predicted path. [Video](https://drive.google.com/file/d/1Ml7AhsIkVRINdJvvRtEchJXQY-KzLrFW/view?usp=drive_link)
> - **Action Execution Failures** – Whether the low-level policy fails even after following the correct path. [Video](https://drive.google.com/file/d/1Gv9UkAiC2tIuvpFkpfgLie8Bhvv-ieiM/view?usp=drive_link)
>
> We observed distinct tendencies across methods:
>
> - For **RVT**, when the VLM prediction is correct:
>   - 72% of failures are due to RVT not following the trajectory.
>   - 28% are due to action execution failures.
>
> - For **3DDA**:
>   - Only 10% of failures are due to not following the trajectory.
>   - 90% are due to action execution failures.
>
> This discrepancy likely arises because RVT includes a re-projection step, complicating trajectory adherence, whereas 3DDA uses a vision tower that processes the original 2D image, simplifying trajectory interpretation.
>
> See [this pie chart](https://drive.google.com/file/d/1HuQIKcd_0_3YBDP5V07W1pL_j585jViP/view?usp=sharing) for a breakdown of failure modes. We have also included this figure in the appendix. Thanks again for the suggestion!
>
>
> **Q: Why human ranking was used as an evaluation metric for the VLM's trajectories? No information about human study details? Why 2D trajectories are not evaluated based on MSE?**
>
> A: Thank you for pointing this out! We have now added more information about the human evaluation to Appendix D.2, restated here:
>
> We use human evaluations to evaluate VLM-produced trajectories instead of MSE against some ground truth trajectory because there are many possible trajectories that can solve any single task. MSE is a metric that assumes only 1 possible ground truth trajectory.
>
> We recruit 4 human evaluators who have not seen the path outputs of HAMSTER before to grade the VLMs based on the instruction: “Provide a rank for each method (1 for best and 4 for worst). In your opinion, which robot trajectory is most likely to succeed. Here is an illustration of the sample to be graded by the human evaluators [link](https://drive.google.com/file/d/13pIZCrVd6qxdPcSrNI3S3-r1klR5c0J5/view?usp=drive_link). Traj goes from blue to red, blue circle means close gripper, red circle means open gripper.'” The evaluators are allowed to give multiple trajectories the same score if they believe those trajectories are tied.

---

> ### Author Response · Authors · 2024-11-20
>
> **Q: Sim experiments results: Only 8% increase in performance, high variance of results?**
>
> A: Thank you for highlighting the issue of high variance. Following your suggestion, we ran the simulated experiments with additional seeds (now 5 in total) to reduce the variance. The updated results show that HAMSTER+3D-DA outperforms 3D-DA in every Colosseum variation category, as shown in the revised Table 2 (reproduced below).
>
> This represents a 31% relative increase in success rate, from 35% to 46%. We believe this improvement is significant, considering the VLM is trained on RLBench but not on Colosseum. As shown in Figure 13 in the appendix, Colosseum scenes are visually distinct from those in RLBench. These results demonstrate that the VLM significantly enhances 3D-DA's ability to generalize to new environments.
>
> |                         | Avg.            | no var          | bac tex         | cam pos         | distractor     | lig col        | man obj col    | man obj siz    |
> |-------------------------|-----------------|-----------------|-----------------|-----------------|----------------|----------------|----------------|----------------|
> | 3D-DA [Ke et al., 2024] | $0.35 \pm 0.04$ | $0.43 \pm 0.06$ | $0.34 \pm 0.07$ | $0.35 \pm 0.11$ | $0.39 \pm 0.11$ | $0.44 \pm 0.13$ | $0.41 \pm 0.04$ | $0.41 \pm 0.11$ |
> | HAMSTER (w 3D-DA)       | $\mathbf{0.46 \pm 0.04}$ | $\mathbf{0.57 \pm 0.03}$ | $\mathbf{0.48 \pm 0.08}$ | $\mathbf{0.39 \pm 0.06}$ | $\mathbf{0.41 \pm 0.05}$ | $\mathbf{0.59 \pm 0.04}$ | $\mathbf{0.57 \pm 0.08}$ | $\mathbf{0.51 \pm 0.10}$ |
>
> |                         | man obj tex     | rec obj col     | rec obj siz     | rec obj tex     | rlb and col    | rlb var        | tab col        | tab tex        |
> |-------------------------|-----------------|-----------------|-----------------|-----------------|----------------|----------------|----------------|----------------|
> | 3D-DA [Ke et al., 2024] | $0.27 \pm 0.04$ | $0.34 \pm 0.10$ | $0.36 \pm 0.05$ | $0.36 \pm 0.12$ | $0.07 \pm 0.03$ | $0.45 \pm 0.12$ | $0.42 \pm 0.06$ | $0.23 \pm 0.04$ |
> | HAMSTER (w 3D-DA)       | $\mathbf{0.48 \pm 0.06}$ | $\mathbf{0.48 \pm 0.05}$ | $\mathbf{0.40 \pm 0.05}$ | $\mathbf{0.56 \pm 0.09}$ | $\mathbf{0.11 \pm 0.10}$ | $\mathbf{0.58 \pm 0.04}$ | $\mathbf{0.56 \pm 0.03}$ | $\mathbf{0.35 \pm 0.07}$ |
>
>
> **Q: How is time encoded in the 2D paths? 2 trajs w/ different lengths? Can this approach encode the speed that the gripper should be moving?**
>
> A: We assign a normalized value between [0, 1] to each point in the 2D trajectory, 0 for the starting point and 1 for the last point. We then scale the red channel with respect to the value for each point. Since the last point is 1 for all trajectories, two trajectories with different length will all end with the same color.
>
> We do not encode the speed at which the gripper should move, however it is a great suggestion for future work!
>
> **Q: Why does the VLM generate 2D trajectories in image space instead of also predicting gripper motion or rotation?**
>
> A: We agree that predicting gripper rotation could provide more information to the low-level action model. However, we chose to predict 2D trajectories and gripper actions for the following reasons:
>
> 1. **Complexity of Gripper Orientation**: The orientation of the gripper is influenced by factors like the spatial relationship between the robot, camera, and table, making it more complex to predict. For example, the gripper pose in camera coordinates varies significantly depending on whether the camera is on the same side or the opposite side of the robot. Predicting 2D paths, which are less affected by such factors, allows the VLM to generalize better across various camera angles in zero-shot settings.
>
> 2. **Cross-Robot Generalization**: To train the VLM with off-domain data from different robots, we require a representation that is transferable across robots. While gripper orientations may vary between robots, the locations of end effectors (2D paths) are more consistent and easier to transfer.
>
> By focusing on 2D trajectories, we ensure better generalization and compatibility across diverse setups and robot types.
>
>
>
> **Q: Why do you think 3D-DA performed worse when the language prompts were removed?**
>
> A: 3D-DA uses a visual attention mechanism that attends CLIP image feature tokens to corresponding CLIP language embeddings. We believe this attention mechanism is important to 3D-DA’s performance; we have updated Appendix B.2 with this discussion.
>
> Let us know if this addresses all of your concerns. Except for the ongoing experiments, we would be happy to address any remaining concerns. Thank you!

---

> ### Author Response · Authors · 2024-11-25
>
> **Q: Why do you only use the front_cam view in RLBench? Can HAMSTER generalize to multi-view?**
>
> Following up on the original response for thie question, now have new experiments demonstrating generalization to a different viewpoint! The experimental environment is shown in this [image](https://drive.google.com/file/d/1O3O7j-cnn49LodFqgpFMJDLpGjFp7y10/view?usp=drive_link), where the **right** camera is the **original** one, and the **left** camera is the **novel** camera. Since the viewpoint robustness is already demonstrated in [video](https://drive.google.com/file/d/1Y6m9GzSjqjOvBt0_timrIPZhLorMapM7/view?usp=drive_link), we focus on evaluating the action-level model here. More specifically, we tested on 6 seen objects and 3 seen containers with 10 different combinations.
>
> The results are summarized in the table below. Both `OpenVLA` and `HAMSTER+RVT2` follow the same setup as in Figure 2 of the paper. `HAMSTER+RVT2 (Concat)` incorporates the 2D trajectory as additional channels, instead of overlaying it on the image, as suggested by Reviewer MHuh. In the metric, `score` is using breakdown point as in Figure 2 and defined in Table 3, while `fully success` only count on trails which are full successful.
>
> From the results, it is evident that both versions of `HAMSTER+RVT2` perform robustly even when the camera setup is changed, **despite the VLM only being trained on `front_cam`**. In contrast, `OpenVLA` exhibits a significant performance drop when tested with a different camera from the one it was fine-tuned on.
>
> We thank the reviewer for this insightful suggestion and will incorporate these findings into the revised manuscript.
>
> | Method                 |       Original| Camera                       | Novel| Camera                       |
> |------------------------|-----------------------|-----------------------|-----------------------|-----------------------|
> |                        | **Score**            | **#Fully Success**    | **Score**            | **#Fully Success**    |
> | OpenVLA               | 6                     | 3                     | 2.25                 | 0                     |
> | HAMSTER+RVT2          | 8.25                  | 7                     | 7.25                 | 4                     |
> | HAMSTER+RVT2 (Concat) | **10**                | **10**                | **9.75**             | **9**                 |

---

> ### Comment · Reviewer_jRfz · 2024-11-26
>
> Thank you for the detailed response and for providing additional experimental results.  The video and quantitative results on novel view generalization are great!  I am more convinced by the simulation results with the extra seeds and error bars included.
>
> I think the human evaluation of 2D trajectories experiment weakens the paper.  I would recommend moving it to the appendix and using that space for the generalization results.  To make the results more compelling, you could add more evaluators or provide details on how the evaluators were instructed/trained to grade results (I am not expecting this in rebuttal period).  If these evaluators were unfamiliar with robotic arms, it's not clear how well they could evaluate the quality 2D projection of a robot trajectory.

---

> ### Author Response · Authors · 2024-11-28
>
> Thanks for the follow-up! We have made changes to the paper based on your suggestions. Detailed response below:
>
> **Q: Move human evals to appendix and put in generalization results?**
>
> Thank you for this suggestion. We have followed this suggestion and now highlight all of the new results in the main paper’s experiments section in the revised rebuttal PDF.
>
> **Q: More details, adding more evaluators, and evaluators familiar with robot arms?**
>
> We have added an additional evaluator for a total of 5 and updated the corresponding table in Appendix Table 5. The evaluators are robot learning researchers who have never seen the predictions of the evaluated VLMs before. Hence, they should be familiar with the quality of robot trajectories. We also list the full instructions given in Appendix D.3, along with all of the above details!
>
> Thank you again for following up. Your suggestions have helped us improve the paper as we have now demonstrated novel view generalization, improved simulation results, highlighted these experiments in the main paper, improved the human evaluations, and updated writing to clarify details! Please let us know if you have any further concerns.

---

> > ### Author Response · Authors · 2024-12-02
> > **HAMSTER Response reminder for reviewer jRFz**
> >
> > Hi reviewer jRFz,
> >
> > As the discussion period comes to a close today, this is a gentle reminder about our most recent response. If there are any pending concerns precluding increasing your score, please let us know.
> >
> > Thank you!

---

### Official Review · Reviewer_MHuh · 2024-11-02

**Soundness:** 3
**Presentation:** 3
**Contribution:** 3
**Rating:** 6
**Confidence:** 3

**Summary:**

The authors in this work propose a hierarchical action model consisting of a high-level vision-language model (VLM) and a low-level imitation learning policy. These two components communicate via a 2D path representation on the image that is predicted by the VLM and used to guide low-level imitation learning. The major advantage of such a design is to enhance the generalization of manipulation tasks by utilizing abundant "off-domain" data for VLM fine-tuning and reducing the requirement of massive amount of "in-domain" on-robot data required by the monolithic action models. Both simulated and real-robot experiments have been conducted to verify the claimed improved generalization performance against the other baselines.

**Strengths:**

- The target problem is important in enhancing the scalability and generalization of robot action models. The proposed idea of using 2D path intermediate representation to bridge the VLMs and low-level action models is neat and sensible;

- The presentation is in general good to follow and clear to understand.

- The idea of using "off-domain" data to fine-tune VLM and consequentially benefit the low-level policy learning is inspiring.

- The experiments include results from both the simulation and the real robot, indicating the workload and applicability of the proposed method.

**Weaknesses:**

- The foremost concern from my perspective is the mismatch between the technical contribution and the way to sell the claimed idea of hierarchical action models. Technically, this work fine-tunes an open-source VLM with easy-to-collect and "off-domain" data for 2-D path/trajectory prediction, which is used to train a low-level imitation learning policy for task execution. The purpose of the authors to enhance the impact of the work is understandable. The current technical and experimental investigation seems insufficient for this purpose. I would suggest the authors either refine the selling points (e.g., "fine-tuning cross-domain VLM for 2D path-guided imitation learning") and switch to more concrete and matched contents for presentation or include more experiments with a special focus on the idea of the hierarchy mechanism itself (detailed below).

- For a more focused investigation of the hierarchy mechanism, if I understand correctly, the first key factor of this idea is the intermediate representation. This key factor has a large impact on the claimed advantage, namely the possibility of utilizing the abundant off-domain training data for fine-tuning. The authors propose to use 2D-path and discuss other alternatives such as 3D-path and key-point representation. It's fine to leave the 3D version for future work. However, the key-point representation can be easily obtained by extracting the start and ending points of the 2-D path. To enhance the focus on the hierarchy mechanism, it would be informative to conduct experiments on this. On the other hand, other related work [1] demonstrates accurate key-point prediction based on open-sourced foundation models without fine-tuning.

- The second key factor in the hierarchy idea is the low-level action models. There are two points on this aspect. The first one is the way to condition the predicted intermediate representation in getting the low-level actions. Though the authors argue that overlapping the 2-D path on the existing images is a good and convenient way, it can help the community to get more insights into this "augmentation" if other ways, like concatenation, can be examined and analyzed both experimentally and empirically. The second one is about the choices of the low-level action model, [1] has demonstrated strong performance without using an imitation learning policy. Why do we need such a policy in this mechanism, which still has to be trained based on a few "in-domain" data? If yes, what's the difference between using a 2D and 3D policy? The investigation done in this work is insufficient.

- There is no explicit empirical evidence for this claim of "drastically reducing the number of demonstrations needed for learning downstream tasks." In real-robot experiments, the better performance after being trained with the same amount of path-augmented images couldn't verify this claim. Having a quantitative experiment on how much data has been saved with the 2D-path guidance would be much more informative. For example, comparing learning curves with different amounts of training data to check if the augmented path can facilitate training efficiency.


[1] Huang, Wenlong, et al. "Rekep: Spatio-temporal reasoning of relational keypoint constraints for robotic manipulation." arXiv preprint arXiv:2409.01652 (2024).

**Questions:**

- Can this intermediate representation limit the versatility of the low-level motions? How do we express the versatility of the motions with the 2D path? On the other hand, the key-point representation can mitigate this constraint. For example, the authors can discuss or empirically evaluate specific types of motions or tasks that might be challenging to represent with 2D paths.

- As mentioned in the experiment part, the number of points of the 2D path can be controlled. How would performance change if we vary the number of points?

- The idea enhances the generalization by using a large VLM for 2D path guidance. Does this increase the computation requirement? Is it worthwhile?

---------------------------------- Post Review ---------------------------------------

I appreciate the efforts of the authors to address my concerns and am pleased to increase my score.

---

> ### Author Response · Authors · 2024-11-20
>
> Thank you for the detailed review! We are happy you find the problem important, presentation clear and the off-domain VLM fine-tuning idea inspiring. We respond to individual points below:
>
> **Q: Additional technical and experimental investigation?**
>
> A: Thank you for your suggestion! We have added discussions on various aspects, including different intermediate representations, methods for providing path information to the low-level action model, the impact of different training demonstrations, and the effect of the number of points on the trajectory. We will update the paper to include the results once the corresponding experiments are completed.
>
> **Q: Focused evaluation of the hierarchy of HAMSTER: Why not just use start and end points instead of the full path?**
>
> A: This is an interesting suggestion! We agree that start and end points can be easily extracted from the 2D-paths, however, they often do not contain sufficient information for specifying the task for the low-level policy. See the two following examples:
> Consider the task of wiping the desk and putting the towel back, see [image](https://drive.google.com/file/d/1TjOGONGfy9glzt3BSDrxQnQfjb4H46DX/view?usp=drive_link). Just the start and end points cannot sufficiently inform what the robot need to do with the towel.
> Similarly, consider the task “Press the yellow button, then blue button and then red button” [image](https://drive.google.com/file/d/12NAz6C0gzmKOn_5ljZPxp-KJtyy0eKdq/view?usp=drive_link). The start and end point alone cannot communicate with the sequence of actions.
> Per your suggestion, we are also running additional experiments with training HAMSTER+3DDA with just the gripper open/close points without drawing any paths. If they finish after the rebuttal, we will include these experiments in the next version of the paper. Thank you!
>
>
> **Q: Comparison to ReKep [1] b/c it doesn’t need a lower level action policy + produces keypoints directly?**
>
> We note that comparing our method to [1] is out of scope, as per ICLR policy. The paper was released to arXiv on September 3, 2024, and hasn’t been published in a peer-reviewed venue. Only works published before July 1, 2024, should be considered for comparison.
>
> Despite this, we compare our method with Rekep: Rekep requires much more state information than imitation learning-based action models. As a model-based method, Rekep needs a TAMP model, a TAMP optimizer, a collision oracle, and a motion planner. It also relies on accurate point tracking for closed-loop optimization. HAMSTER does not require any of these.
>
> For key-point prediction, Rekep runs K-means clustering on DINOv2 features to get keypoint proposals and selects relevant ones with VLM. This approach depends on the quality of DINOv2 features and hyperparameters for K-means clustering, so it likely struggles in cluttered environments like [video](https://drive.google.com/file/d/1uxNvWj5IuTQ9sSw4BVC9Z1Bbi7MlCkTv/view?usp=drive_link). We’re currently deploying ReKep in our real-world experiments for quantitative comparison; we’ll share updates as the experiments progress and include a full comparison in the camera ready.
>
> [1] Huang, Wenlong, et al. "Rekep: Spatio-temporal reasoning of relational keypoint constraints for robotic manipulation." arXiv preprint arXiv:2409.01652 (2024).
>
> **Q: Why draw paths on images? Compare against other methods of giving path information to the low-level policy?**
>
> A: We draw VLM-produced paths onto the image for the lower-level policy to condition on because it is a **generalizable** method of conditioning the lower level policy on paths that works for any image input policy.
>
> Per your suggestion, we are now running **additional experiments** comparing against a version of HAMSTER+RVT2 which concatenates just the path image with the RGB input image (6 channel input) to specify the path. Note that this already is less generalizable across lower-level policies, as we cannot easily perform this with 3D-DA due to it using pre-trained CLIP image encoder backbones that take 3-channel RGB images as input.
>
> **Q: No experiments showing “drastically reduced number of needed demonstrations”?**
>
> A: Thanks for pointing this out. We agree that in the related work section (Sec. 2), we mentioned our methods would “drastically reduced number of demonstrations” without explicit experiments to support it. Since this is not the primary focus of the paper, we have modified this sentence to  better reflect the focus of the paper, that “Our work takes a hierarchical approach, enabling us to use specialist lower-level policies that take in additional inputs the VLMs cannot support, such as 3D pointclouds, to enable better imitation learning.”
>
> Further, we agree with your suggested experiment. Hence, we are now running additional experiments in which we train HAMSTER+3D-DA on only 50% of the data to compare it with vanilla 3D-DA. We will add those results in the next version of the paper as soon as they are finished.

---

> > ### Author Response · Authors · 2024-11-20
> >
> > **Q: Does the 2D path representation limit the versatility of the low-level motions?**
> >
> > A: As shown in Figure 2, our low-level action model receives inputs similar to its original version: an RGB-D image with the 2D path overlaid and a simplified language instruction specifying the task. Given these inputs, we believe the low-level model retains its versatility. Additionally, as shown in Figure 3, providing 2D paths allows models like RVT2 and 3DDA to perform better on basic tasks that do not require generalization. We hypothesize that this improvement arises from partially reducing the perception burden, allowing the model to better focus on action execution.
> >
> > **Q: The number of points of the 2D path can be controlled. How would performance change if we vary the number of points?**
> >
> > A: This is a great suggestion; we are running these experiments but unsure if they will finish by the rebuttal period’s end, as this requires retraining the VLM as well as re-doing the experiments with the low-level policy. Regardless, if accepted we will add this analysis to the camera ready version of the paper.
> >
> > **Q: The idea enhances the generalization by using a large VLM for 2D path guidance. Does this increase the computation requirement? Is it worthwhile?**
> >
> > A: Although we do increase the training and inference computation compared with small imitation learning models, it is far less computation compared to monolithic VLA models that call upon the large model at every inference step. We only perform 1 forward pass of the VLM at the beginning of the rollout. Our results in Fig 3 demonstrate:
> > With less computation, HAMSTER achieves up to **4X** higher success rate compared with OpenVLA on various generalization evaluations.
> > Even with little generalization required (“basic” column in Fig 3) , HAMSTER can significantly increase the small imitation learning policy’s success rate by **1.5X**.
> >
> > Let us know if this addresses all of your concerns. Except for the ongoing experiments, we would be happy to address any remaining concerts. Thank you!

---

> ### Author Response · Authors · 2024-11-27
>
> We have now added **new experiments** addressing pending concerns above! We list them below and have **updated the paper with these results in Appendix D.3.**
>
> **Q: Why draw paths on images? Compare against other methods of giving path information to the low-level policy?**
> We now have new experiments in the real world giving path information to RVT2 by concatenating the path image as 3 additional channels in addition to the RGB image channels. Here we conducted 10 pick-and-place trials with 6 seen objects and 3 seen containers, and also with novel camera viewpoints.  The results are as shown below, where column 4 (HAMSTER+RVT2) is the one used in Figure 2 which draws a path directly on the image, while column 5 (HAMSTER+RVT2 (Concat)) is the new experiment.
> We can see that although drawing directly on the image is a more generalizable way for different low-level action models, HAMSTER with concatenated images works even better with RVT2. Thank you for the suggestion!
>
> | Method                 | Original Camera       |                       | Novel Camera          |                       |
> |------------------------|-----------------------|-----------------------|-----------------------|-----------------------|
> |                        | **Score**            | **#Fully Success**    | **Score**            | **#Fully Success**    |
> | OpenVLA               | 6                     | 3                     | 2.25                 | 0                     |
> | HAMSTER+RVT2          | 8.25                  | 7                     | 7.25                 | 4                     |
> | HAMSTER+RVT2 (Concat) | **10**                | **10**                | **9.75**             | **9**                 |
>
> **Q: Additional technical and experimental investigation?**
>
> The above experiment table now also demonstrates an additional experiment investigating the hierarchy of the model as suggested. We demonstrate that HAMSTER is much more camera-view invariant than OpenVLA. This specifically highlights the benefits of hierarchy; HAMSTER’s VLM is trained on additional web pixel point data containing images of different scenes w/ diverse views, and RVT2 is a specialized imitation learning policy which performs camera view reprojection and is more robust to novel views.
>
> **Q: No experiments showing “drastically reduced number of needed demonstrations”?**
> Finally, we have also added **new experiments** demonstrating that HAMSTER can perform better than the underlying imitation learning model even with far less demonstrations. Our results, obtained over 5 seeds across 5 short-horizon tasks in Colosseum (`slide_block_to_target`, `place_wine_at_rack_location`, `insert_onto_square_peg`, `stack_cups`, `setup_chess`) shown in the table below, demonstrate that with just 50% of the data, HAMSTER+3D-DA performs **2x** better than 3D-DA with 100% of the data.
>
> | Model              | Success Rate |
> |------------------------|---------------------|
> | HAMSTER+3D-DA              | 0.43           |
> | HAMSTER+3D-DA (50% Data)    | 0.36           |
> | 3D-DA Naive                   | 0.18           |
>
>
>
> Please let us know if you have any remaining concerns. Thank you!

---

> > ### Comment · Reviewer_MHuh · 2024-11-27
> > **Response from Reviewer MHuh**
> >
> > Thank you for the additional experiments. I appreciate the efforts on this and recommend the authors to include these results into the main draft. Regarding the comparison with Rekap, it would be nice to have, but not necessary, the key message here is to compare with the way of using the key-points as the intermediate representation. This comparison is still missing. Regarding the versatility of the low-level action models, I apologize for the ambiguous term; what I actually want to mean is the impact of adding a specific path on the performance of modeling the **multi-modal** action distributions. For example, there might be multiple ways to execute the tasks. By learning with the 2D path guidance, is the model still able to learn the multi-modal distribution? If not, then maybe a very simple policy architecture with MLP as the action head is sufficient to achieve good performance.

---

> ### Author Response · Authors · 2024-12-02
>
> Thanks for the followup! We have now addressed your two points on multi-modality and a Rekep comparison:
>
> **Q: Specific path’s effects on multi-modal action distributions?**
>
> Thanks for the clarification about your original question. We have run **new experiments** demonstrating that yes, indeed HAMSTER’s low-level policy is able to learn a multi-modal distribution! We evaluated HAMSTER+RVT2 with the same path drawn on the image (**see the setup [here](https://drive.google.com/file/d/1pCBUZlmiLh3UtN6cZebmYl2BZmef9Om1/view?usp=drive_link)**) but with different objects to place and receptacles of varied sizes and heights. This RVT2 policy is able to vary the low-level actions to change the height at which it picks up and places the objects despite being conditioned on the same path: see video 1 [grape in bowl](https://drive.google.com/file/d/1GGZsp7x2mKJwMVpBYXTcUXK8zgIgWXuG/view?usp=drive_link), video 2 [milk in bowl](https://drive.google.com/file/d/1wWJzbKidGjojt5iElnydnRETYOe8pr6e/view?usp=drive_link), video 3 [grape in mug](https://drive.google.com/file/d/1GThcyArO-ABhE6YP35_p1PT2mvdCQSF3/view?usp=drive_link)
>
>
> HAMSTER is able to do this because (1) the 2D path still underspecifies the low-level environment actions in terms of depth and rotation, and (2) we explicitly train the low-level imitation learning policy with noisy ground truth paths so it does not overfit to following the exact action sequence outlined by the path.
>
> If our paper is accepted, we will add these experiments to the camera ready. Thank you for the great suggestion that allowed us to demonstrate that HAMSTER can produce multi-modal action distributions!
>
>
>
>
> **Q: Comparison to keypoint based methods like ReKep b/c it doesn’t need a lower level action policy + produces keypoints directly?**
>
> In our previous discussion, we addressed the question of why a low-level policy eliminates the need for the state information that ReKep requires. In short, the difference lies in the contrast between model-based TAMP systems and model-free learning-based methods.
>
> Now, we also address why fine-tuning the model is essential, even when ReKep demonstrates high accuracy in its benchmarks. We evaluated ReKep using their official repository, utilizing the same input image and the same bounded range of point cloud data. In one test scenario, we tasked the network with predicting a trajectory to press four buttons within the same image.
>
> The results are visualized **[here](https://drive.google.com/file/d/13nMhW_2wYtSOoS2M9a-SIThcy6KUzGag/view?usp=drive_link)**,where the leftmost column is the input to each VLM in HAMSTER and ReKep, the right 4 columns are their prediction under different tasks.
>
> In the results image, you can see that ReKep’s pipeline misidentifies where the button is in 3 out of 4 of the tasks, while HAMSTER correctly identifies 4/4 times. It’s important to ensure the keypoints for the buttons are correctly placed on top of the buttons to be able to solve button-related tasks here. Without correct keypoints, and without a low-level policy that can learn to correct for these keypoint identification errors, ReKep would fail at executing these tasks.
>
> What’s more, even for the successful case, ReKep proposes a constraint which presses the gripper 5 cm below the keypoint. However, the button is only 3 cm thick, therefore this operation could harm the robot.
>
> These results highlight the strengths of HAMSTER vs ReKep in that it does not require state information (through use of a low-level policy) and our use of a fine-tuned VLM for path prediction results in accurate object-centric paths.
>
> We detail the ReKep output results below:
> [press the yellow button](https://drive.google.com/file/d/194vIm77hMbF2_sHtFa6lir-B_4Dw26Sl/view?usp=drive_link0),
> [press the blue button](https://drive.google.com/file/d/1cNxmKmhUK22T-Luc-MYbX1ay7Yzn8N9j/view?usp=drive_link),
> [press the button with color of fire](https://drive.google.com/file/d/1BNVyeKdJA9FpYaQnY_mDh7ZDw-ZQzDfS/view?usp=drive_link),
> [press button with color of leaf](https://drive.google.com/file/d/16w8Mvw-OphkCvQ1vbP-hVCMOXIOpGv5r/view?usp=drive_link).
>
> Please let us know if you have any further concerns preventing a higher score. Thank you!

---

> > ### Comment · Reviewer_MHuh · 2024-12-02
> > **Response from Reviewer MHuh**
> >
> > Thank you again for the additional experiments. The experiment results are quite informative but still don't resolve my concern.
> > - Regarding the multi-modal action distribution modeling, in the additional experiment, though with different objects to place and receptacles of varied sizes and heights, the predicted actions don't seem to change much, still roughly following the guided path. A more helpful experiment to investigate this would be compare the diversity of multiple predicted feasible roll-outs between the policy trained with the path-guidance and the one without. The concern behind is to understand whether the path-guidance imposes additional constraints on the policy training. I believe this will be informative for both practitioners and future researchers.
> > - For the comparison with Rekap, I appreciate the added experiments to show the benefits of the fine-tuned VLM on key-points/path detection, which is obvious though. Therefore these results don't address my concern regarding the impacts of training low-level action models with different intermediate representations, i.e., the key-points vs. the 2D-path.

---

> ### Author Response · Authors · 2024-12-02
> **Response regarding keypoints v.s. paths and multimodal ability for low-level model**
>
> Thanks for the quick response. Following we address both your concerns:
>
> **Q: Impacts of training low-level action models with different intermediate representations, i.e., the key-points vs. the 2D-path?**
>
> We have previously demonstrated through examples that just keypoints extracted from start/end points or gripper open/close points are insufficient in solving manipulation tasks, particularly long-horizon ones, such as [wipe the table and put it back](https://drive.google.com/file/d/1TjOGONGfy9glzt3BSDrxQnQfjb4H46DX/view?usp=drive_link) and [press green, blue red button in order](https://drive.google.com/file/d/12NAz6C0gzmKOn_5ljZPxp-KJtyy0eKdq/view?usp=drive_link).
>
> Further, inspired by the reviewer’s question, we investigate a stronger keypoint representation: drawing VLM-predicted path keypoints without drawing the lines in between.We find that both qualitatively and quantitatively that the 2D paths are more effective than just keypoints.
>
> We agree with the reviewer that for some tasks keypoints may be sufficient but it need be the case as shown below. For example, consider the task of `stack cups’, where it is hard to understand distinguish the path with just the keypoints as shown here: [path image](https://drive.google.com/file/d/152HMz3VjKKLsmuV4fHJ33CaI_Me0rV6p/view?usp=drive_link)  vs. this [keypoints image](https://drive.google.com/file/d/12PGtGPFiEMZlxX4Poye_EywSF7ThlCKX/view?usp=drive_link)
>
> This observation is also verified empirically as shown below. Over 5 seeds in Colosseum, we find that just drawing keypoints performs worse than drawing full trajectories; however both strategies are better than just the low-level policy alone. This outcome may be due to the fact that providing complete 2D paths, rather than isolated keypoints, makes it easier for the low-level action model to interpret and follow the guidance effectively.
>
> | Model                          | Stack Cups |
> |--------------------------------|------------|
> | HAMSTER+3D-DA (2D Paths)                 | **0.63**   |
> | HAMSTER+3D-DA (Keypoints) | 0.44       |
> | 3D-DA                    | 0.04       |
>
> **Q: Multimodal experiment: still following path? Does path-guidance impose additional constraints?**
>
> We clarify that we actually explicitly aim to constrain the policy with path guidance. As we have demonstrated experimentally, this helps the policy generalize to new tasks with less data, (Figure 3). This path constraint is reasonable as our fine-tuned VLM is able to produce high quality paths (see Table 5 and multi-view consistency [video](https://drive.google.com/file/d/1nCbc2n2wLjgN0bKN9Rwe7RzCE0uFgGbY/view?usp=sharing)).
>
> Therefore, the multi-modal experiments in our previous response successfully demonstrate that the policy is still following the path in 2D space, but able to produce multi-modal low-level actions by adapting to the heights of various objects.
>
> However, we agree with the previous suggestion that “maybe a very simple policy architecture with MLP as the action head is sufficient.” This actually highlights one of the strengths of HAMSTER: that it is not tied to a specific low-level imitation learning architecture. We apologize for the previous misunderstanding regarding these experiments: we will later include experiments in the camera-ready with a simple Gaussian MLP policy head.
>
> Please let us know if this addresses your concern about path guidance constraints.
>
> We are happy to discuss further and attempt any suggested experiments in the remaining discussion period.

---

> > ### Author Response · Authors · 2024-12-04
> > **Response regarding to impact of number of points on paths**
> >
> > Thank you for your recognition! Due to the extension of the discussion period, we managed to get preliminary results for the question regarding to varied number of points of the 2D path
> >
> > **Q: The number of points of the 2D path can be controlled. How would performance change if we vary the number of points?**
> >
> > To investigate the effect of the number of points on the 2D path, we train the VLM to predict 1. paths simplified using RDP algorithm, which simplify paths in short horizon tasks to **3-5 points** and is what we used in the paper. We denote these paths as **RDP** in the following; 2. Paths represented with 20 points sampled on the path with same step size, denoted as **20p** in the following. We keep points where the gripper is executing operation of open or close in both methods.
> >
> > We train the network on RLBench 80 tasks with 1000 episodes for each task and test it on 25 episodes on the task of **close jar**. We tried both VILA1.5-3B (denoted as **3B**) and VILA1.5-13B (denoted as **13B**) as our backbone. Thus we have in total 4 combinations over 2 backbones and 2 designs of path representations.
> >
> > We visualize the result in this [image](https://drive.google.com/file/d/1F2nLpNKOk0UsS1xMr6gSpw-dsmxXIkx2/view?usp=drive_link) . More results can be found in this [folder](https://drive.google.com/drive/folders/1Sf3PxrmZygn2T8PfCudNjV4jFfNFkmFL?usp=drive_link). The task **close jar** is to pick up the lid and close it on the correct jar, as illustrated in this [video](https://drive.google.com/file/d/1IMGl_-O65yHw84b6c8S-2vm-9oTBMv75/view?usp=drive_link).
> >
> > From this result we can see that when using smaller models, like VILA1.5-3B, paths represented using points extracted using RDP algorithm outperforms paths represented with a fixed number of 20 points significantly. When the network becomes larger to the level of 13B, the VLM is able to handle the representation using 20 points and both two path representations work perfectly. We believe that is because when points are simplified using the RDP algorithm, we usually need less points to represent the path and helps the model to pay more attention to predict the accurate position for the gripper open/close points.
> >
> > We thank the reviewer for this insightful point. We will further investigate it on real images to see whether this observation will happen on 13B when input images and tasks become more challenging in the real world. If the paper is accepted, we will include a detailed follow-up analysis in the revised version.

---

### Official Review · Reviewer_Wx1Z · 2024-11-04

**Soundness:** 4
**Presentation:** 3
**Contribution:** 3
**Rating:** 8
**Confidence:** 4

**Summary:**

The proposed method, as shown in Fig.1 and 2, leverages the advantages of imitation learning and VLA models to increase generalization ability. It distills knowledge of the task by visual path representation with VLA and uses dataset/task-specific models to deal with the lower-level controls.

**Strengths:**

1. The baseline comparisons are fair and detailed. More convincing with human evaluation.
2. As stated in the summary, HAMSTER leverages the advantages of imitation learning and VLA models to increase generalization ability. It distills knowledge of the task by visual path representation with VLA and uses dataset/task-specific models to deal with the lower-level controls.

**Weaknesses:**

1. Notations of proprioception are used differently (s and p, line 336 and 234). It confuses and breaks the whole problem statement.
2. There are no (a), (b), and (c) Fig. 5.
3. The system still requires human-engineered prompts for the VLM and other efforts such as "Conditioning on Paths" (page 7) and running the Ramer-Douglas-Peucker algorithm.

Minor:
1. missing period in line 215.
2. i.e -> i.e., (line 231).

**Questions:**

1. What is the use of the intermediate representations $l_i$ of the hierarchical VLM?
2. In the fine-tuning phase, are the outputs of VLM normalized across datasets? For instance, the pixel locations are normalized to $[0, 1]$ or not?
3. What is the length of the output of the Ramer-Douglas-Peucker algorithm?
4. Which part of the system has the 3D knowledge of the world? Does VLA project 3D paths in the real world to images or does the policy deal with it?
5. Please unify the notations in the problem formulation, e.g., proprioception vs. path-labeler and proprioception vs. states. Formally define and explain why they share the same notations if you have to use them repeatedly.

---

> ### Author Response · Authors · 2024-11-20
>
> Thank you for the positive feedback! We are happy you find the paper sound, our evaluation convincing, and the baseline comparisons fair and detailed. We respond to individual concerns below:
>
> **Q: Unify notations in the problem formulation.**
>
> Thanks for the feedback, we have now fixed these issues!
>
> **Q: What is the use of the intermediate representations li  of the hierarchical VLM?**
>
> A:$l_i$ is a typo that instead should refer to the path produced by the VLM, $p_i$. We have fixed this typo in Section 4.1.
>
> **Q: In the fine-tuning phase, are the outputs of VLM normalized across datasets? For instance, the pixel locations are normalized to [0,1] or not?**
>
> A: Yes, we normalized the pixel location to [0, 1] with respect to the height and width of the image. We have clarified this in Section 4.1.
>
> **Q: What is the length of the output of the Ramer-Douglas-Peucker algorithm?**
>
> A: We do not limit the length of the RDP algorithm. The output length is typically between 2-5 points for each short horizon task. We have added this detail to appendix Section B.
>
> **Q: Which part of the system has the 3D knowledge of the world? Does VLA project 3D paths in the real world to images or does the policy deal with it?**
>
> A: The low-level action model handles the 3D understanding and predicts accurate actions. After the VLM predicts 2D paths, we draw these 2D paths on the image. The low-level action model takes this updated rgb image along with the point cloud as input to predict actions in 3D.
>
> **Q: Minor typos**
>
> Thank you! These are fixed.
>
> Let us know if this addresses all of your concerns. We would be happy to address any concerns that still remain. Thank you!

---

> > ### Comment · Reviewer_Wx1Z · 2024-11-20
> >
> > Thanks for the reply. Following the last question, I wonder what would happen if you test the HAMSTER pipeline on a new task with a tilted camera angle. I ask this because, in such cases, the low-level controller needs to interpret pixels from the RGB inputs to the depth in the real world.

---

> > > ### Author Response · Authors · 2024-11-23
> > >
> > > We conducted an experiment to evaluate the performance of our methods under a different camera setup. The experimental environment is shown in this [image](https://drive.google.com/file/d/1O3O7j-cnn49LodFqgpFMJDLpGjFp7y10/view?usp=drive_link), where the **right** camera is the **original** one, and the **left** camera is the **novel** camera. Since the viewpoint robustness is already demonstrated in [video](https://drive.google.com/file/d/1Y6m9GzSjqjOvBt0_timrIPZhLorMapM7/view?usp=drive_link), we focus on evaluating the action-level model here. More specifically, we  tested on 6 seen objects and 3 seen containers with 10 different combinations.
> > >
> > > The results are summarized in the table below. Both `OpenVLA` and `HAMSTER+RVT2` follow the same setup as in Figure 2 of the paper. `HAMSTER+RVT2 (Concat)` incorporates the 2D trajectory as additional channels, instead of overlaying it on the image, as suggested by Reviewer MHuh. In the metric, `score` is using breakdown point as in Figure 2 and defined in Table 3, while `fully success` only count on trails which are full successful.
> > >
> > > From the results, it is evident that both versions of `HAMSTER+RVT2` perform robustly even when the camera setup is changed. In contrast, `OpenVLA` exhibits a significant performance drop when tested with a different camera from the one it was fine-tuned on.
> > >
> > > We sincerely thank the reviewer for this insightful suggestion and will incorporate these findings into the revised manuscript.
> > >
> > > | Method                 |       Original| Camera                       | Novel| Camera                       |
> > > |------------------------|-----------------------|-----------------------|-----------------------|-----------------------|
> > > |                        | **Score**            | **#Fully Success**    | **Score**            | **#Fully Success**    |
> > > | OpenVLA               | 6                     | 3                     | 2.25                 | 0                     |
> > > | HAMSTER+RVT2          | 8.25                  | 7                     | 7.25                 | 4                     |
> > > | HAMSTER+RVT2 (Concat) | **10**                | **10**                | **9.75**             | **9**                 |

---

> > > > ### Comment · Reviewer_Wx1Z · 2024-11-27
> > > >
> > > > Thanks for the reply. I appreciate the efforts on the additional experiments. I will maintain my positive score for now.
> > > >
> > > > As a side note, I do not think the following has been addressed: "requires human-engineered prompts for the VLM and other efforts such as "Conditioning on Paths" (page 7)". Please correct me if I am wrong.

---

> ### Author Response · Authors · 2024-11-21
>
> Thank you for the insightful question! The high-level VLM is expected to handle new camera viewpoints well, as demonstrated in this [video](https://drive.google.com/file/d/1Y6m9GzSjqjOvBt0_timrIPZhLorMapM7/view?usp=drive_link). For the low-level controller, we would expect a 3D policy to generalize better than a 2D one since it's lifted to a 3D space in a common frame of reference rather than being strictly viewpoint dependent. We are currently running experiments to validate this, we will report back soon.

---

> ### Author Response · Authors · 2024-12-02
> **Response regarding HAMSTER prompts and path conditioning human effort**
>
> Thanks for the followup! We respond to each point below:
>
> **Q: Human-engineered prompts for the VLMs?**
>
> We clarify that our prompt is quite short—in fact, our RT-Trajectory baseline prompt in Fig 10 is significantly longer than ours in Fig 8—and also not hand-engineered for our experiments: we use the same prompt template that was used for fine-tuning the VLM on path prediction for all HAMSTER evaluations in simulation and real.
>
> In fact, even after training, the VLM is quite robust to prompt variations!
> We have shown results in **[this google doc](https://docs.google.com/document/d/1UXp3GzCpuDX0Yq5eK93yad8avbbJ2_RAD2wlYRLR08c/edit?usp=sharing)** where we use a screenshot from OpenVLA of the **coke on Taylor Swift** task and use varied questions and prompt templates to query the VLM. We bold the words which are changed over the original prompt template. Benefiting from the strong ability from the original VLM, it can understand different ways describing the same task and generalize to different prompt templates.
>
>
> **Q: Human effort: Conditioning on paths? Running the Ramer-Douglas-Peucker algorithm?**
>
> We believe this approach to be quite general that does not require any scenario-specific engineering. We draw paths directly on the images in the same way no matter the low-level imitation policy or task, both in simulation and in real world experiments. We also run the RDP algorithm with the same parameter ($\epsilon=0.05$) for all data as a simple preprocessing step before training the VLM. We did not tune this parameter, we simply selected it at the beginning as it is the default parameter and the paths simplified under this param look reasonable.
>
> If accepted, we will add the above important discussion to the appendix for the camera ready. Please let us know if you have any further concerns precluding a higher score. Thank you!

---

> > ### Comment · Reviewer_Wx1Z · 2024-12-02
> >
> > Thanks, I am satisfied with the response. I will raise my score.

---

### Official Review · Reviewer_95o8 · 2024-11-07

**Soundness:** 2
**Presentation:** 3
**Contribution:** 2
**Rating:** 5
**Confidence:** 3

**Summary:**

This paper introduces a hierarchical model for robotic manipulation using Vision-Language Models (VLMs) to generate intermediate 2D waypoints, which then guide a low-level control policy. The idea is to leverage high-level VLM-based planning to produce simple 2D paths, using less specific and more accessible data sources like videos or simulations. This approach aims to make manipulation models more generalizable across different environments without requiring extensive, robot-specific training data.

**Strengths:**

- The hierarchical approach has a nice structure, separating high-level VLM predictions from low-level control, which could allow for integration of varied data sources in future applications.
- The focus on using general, “cheap” data sources is interesting and relevant, as scaling data collection in robotics is always a challenge. This could be an efficient way to sidestep the need for large, domain-specific datasets.
- The generalization goal aligns well with current interests in making robotic systems more adaptable to diverse environments and tasks, and this modular setup has potential.

**Weaknesses:**

- The paper doesn’t seem to address some core challenges in using VLMs for manipulation, such as understanding 3D spatial relationships. VLMs trained on language and images aren’t inherently equipped for the depth of 3D reasoning required for more complex manipulation.
- The tasks used are relatively simplistic and limited mostly to pick-and-place scenarios, which doesn’t fully showcase the potential of this approach. Although these tasks have merit, it’s unclear how well the model would handle more challenging or continuous tasks that require a stronger 3D understanding.
- In terms of novelty, the approach doesn’t seem to introduce significant improvements over existing methods for VLM-based waypoint generation. While the structure is clear and modular, there’s limited demonstration that it extends much beyond what current methods already accomplish.
- A primary limitation is that the model relies only on 2D waypoints generated from single-view images, which restricts the range of tasks it can realistically handle. For tasks that require nuanced 3D spatial awareness, like pushing or rotating objects, the 2D-only input may limit its applicability.
- While the system’s hierarchical design has promise, the 2D waypoint restriction means it may not leverage the full potential of VLMs in manipulation tasks, and this makes it less flexible for more sophisticated, real-world interactions.

**Questions:**

- How would the model perform on tasks requiring intricate 3D manipulation, like pushing or rotating objects? Given its current reliance on 2D waypoints, would extending this approach to more complex, spatially-aware tasks be feasible?
- Why were primarily pick-and-place tasks chosen? It would be insightful to see how the model handles a more varied set of challenges to get a clearer sense of its adaptability.
- Since VLM-based waypoint generation has been explored in past work, could the authors clarify how this approach differs from or improves upon those existing methods?
- Are there future plans to incorporate 3D input or multi-view data into the model? This could help address some of the limitations around spatial understanding and potentially broaden the approach’s applicability.

---

> ### Author Response · Authors · 2024-11-20
>
> We thank you for the thoughtful feedback. We are happy you find the paper interesting and like the hierarchical approach. We respond to individual comments below:
>
>
> **Q: Only works on pick and place tasks?**
>
> A: We include more than just pick and place tasks (see appendix Table 3 for the full task list). We have additionally grouped tasks by their type below:
>
> | Task           | openvla | HAMSTER+RVT2 | HAMSTER+3DDA |
> |----------------|---------|--------------|--------------|
> | pick and place | 0.46    | **0.79**         | 0.78         |
> | press button   | 0.25    | 0.50         | **0.63**         |
> | knock down     | 0.41    | 0.47         | **0.66**         |
>
>
> We can see HAMSTER outperforms OpenVLA across all task groups and is therefore not limited by the VLM’s 2D perception capabilities in solving more complex tasks. For clarity we have now added this as Table 4 in the appendix and reference it in Section 5.1.
>
> **Q: No pushing/rotation tasks due to VLM’s 2D perception?**
>
> A: The aforementioned “knock down” task in the table above involves pushing an object down. We also demonstrate the ability of HAMSTER’s VLM to generate rotations and pushing in the following video [stow object in shelf](https://drive.google.com/file/d/1RBl16Cr0MnKk_HdU3jXqYir90j9sK9Mq/view?usp=drive_link),  [close drawer](https://drive.google.com/file/d/1Q8tpuzJhU3kDbKVxgOivq2A7FjDvdT-Q/view?usp=drive_link). Due to time constraints, we don’t have full task evaluations for rotation tasks but we are running these experiments for the future version of the paper.
>
> **Q: You’re not solving 3D perception with VLMs? 2D only is a limitation?**
>
> A: The 3D information can be used for two purposes: reasoning and understanding the task, and accurately predicting the action. Our VLM, although limited by monocular RGB input, can still address many 3D reasoning challenges, such as inferring the top, middle, or bottom drawer as shown in [this image](https://drive.google.com/file/d/1XIKYsIZXh73W0PQy171JhIsJDt-dASyD/view?usp=drive_link). For action prediction, we leverage an action model that incorporates 3D input, enabling us to handle more complex tasks effectively.
>
> **Q: Novelty — comparison against prior work on VLM keypoint methods?”**
>
> A: The two most similar VLM waypoint generation works would be Code-as-policy in RT-Trajectory [1] and LLaRVA [2]. RT-trajectory introduces waypoint generation as a method of specifying tasks to policies; in contrast, **our novelty** comes from a pipeline for using cheap, cross-modal data to fine-tune VLMs to produce accurate paths for 3D imitation learning policies. We directly compare with RT-Trajectory in Table 2, replicated below:
>
> | Method    | VLM       | Finetuning | Data              | Rank Exc. Real RLB. | Rank Real RLB. | Rank All |
> |-----------|-----------|------------|-------------------|---------------------|----------------|----------|
> | RT-Traj.  | 0-shot GPT-4o | -          | 3.40              | 3.63                | 3.52           |          |
> | RT-Traj.  | CaP GPT-4o | -          | 3.57              | 3.36                | 3.46           |          |
> | HAMSTER   | VILA      | Our Exc. Sim RLB. | 1.78          | 2.39                | 2.11           |          |
> | HAMSTER  | VILA      | Our        | **1.59**          | **1.28**            | **1.42**       |          |
>
> Where we see that HAMSTER’s VLM is best-ranked by humans for accurate path predictions, demonstrating the need for fine-tuning on cross-modal data. LLaRVA, on the other hand, uses waypoint prediction as an auxiliary objective to fine-tune a monolothic VLA; our experiments in Figure 3 demonstrate that our hierarchical VLA approach is superior to monolothic ones (OpenVLA). More detailed information can be found in Section D in Appendix.
>
> [1] Gu et al., RT-Trajectory: Robotic Task Generalization via Hindsight Trajectory Sketches, 2024.
> [2] Niu et al.,  LLARVA: Vision-Action Instruction Tuning Enhances Robot Learning, 2024.
>
> **Q: Future plans incorporating 3D/multi-view data into the VLM?**
>
> A: We appreciate the reviewer’s suggestion! In future work, we plan on incorporating multi-view input to the VLM and fine-tuning the VLM on 3D data for effective high-level even in multi-view robot setups.
>
> Let us know if this addresses all of your concerns. Except for the ongoing experiments, we would be happy to address any remaining concerns. Thank you!

---

> > ### Author Response · Authors · 2024-11-27
> >
> > Dear Reviewer 95o8,
> >
> > We sincerely thank you for the time and effort you have devoted to reviewing our paper. As the rebuttal discussion period nears its conclusion, we kindly ask you to review our submitted rebuttal. We have carefully addressed your comments and greatly value your feedback. Should you have any additional questions or suggestions, we would be more than happy to address them.
> >
> > Thank you again for your thoughtful review and constructive feedback.

---

> > > ### Author Response · Authors · 2024-12-02
> > > **HAMSTER can perform 3D reasoning with 2D VLMs**
> > >
> > > Hi reviewer 95o8:
> > >
> > > We now have new experiments demonstrating the ability of HAMSTER to perform 3D reasoning even with 2D paths.
> > > We evaluated HAMSTER+RVT2 with the same path drawn on the image (**see the setup [here](https://drive.google.com/file/d/1pCBUZlmiLh3UtN6cZebmYl2BZmef9Om1/view?usp=drive_link)**) but with different objects to place and receptacles of varied sizes and heights. The RVT2 policy is able to vary the low-level actions to change the height at which it picks up and places the objects despite being conditioned on the same path: video 1 [grape in bowl](https://drive.google.com/file/d/1GGZsp7x2mKJwMVpBYXTcUXK8zgIgWXuG/view?usp=drive_link), video 2 [milk in bowl](https://drive.google.com/file/d/1wWJzbKidGjojt5iElnydnRETYOe8pr6e/view?usp=drive_link), video 3 [grape in mug](https://drive.google.com/file/d/1GThcyArO-ABhE6YP35_p1PT2mvdCQSF3/view?usp=drive_link).
> > >
> > > HAMSTER is able to do this 3D reasoning because we can use powerful low-level imitation learning policies which are able to perform 3D reasoning even though they are trained on 2D paths.
> > >
> > > If our paper is accepted, we will add these experiments to the camera ready. Please let us know if there are any other concerns precluding increasing your score. Thank you!

---

### Author Response · Authors · 2024-12-02
**HAMSTER General Response: New experiments and paper updates**

R1: 95o8, R2: Wx1Z, R3: MHuh, R4: jRfz

We thank the reviewers for their thoughtful feedback and engagement during the discussion period. We are happy the reviewers like that our hierarchical VLA approach allows for scalable use of cheap, off-domain data (R1, R2, R3, R4), that our experiments are detailed (R2, R4), and the presentation is good (R1, R2, R3, R4).

We’ve addressed each reviewer’s concerns individually. Following their suggestions, we have added the following major new experiments and paper changes:

- **New experiments** demonstrating HAMSTER **generalizing to new views** (Table 2, [video](https://drive.google.com/file/d/1nCbc2n2wLjgN0bKN9Rwe7RzCE0uFgGbY/view?usp=sharing)) with a **4.3X** task progress improvement over OpenVLA (R2, R3, R4)
- **Real-world evals** with HAMSTER+3D-DA (Figure 3), an additional imitation learning method (R4), split results by type to show **HAMSTER is better on non pick-place tasks** (Table 6), like pressing buttons and knocking down objects (R1, R4)
- Detailed **failure mode analysis** to Appendix E and Figure 13 (R4).
- **New experiments** demonstrating HAMSTER+3D-DA works **2X** better than standard 3D-DA with just 50% of the data (Table 3) (R3)
- **New experiments** demonstrating that HAMSTER’s modularity allows the low-level action model to work effectively with other path input representations (Table 2) (R3).

As the discussion period is coming to a close, we gently remind reviewers to follow-up on our latest responses as we have addressed all pending concerns. Thank you!

---

### Meta-Review · Area_Chair_x19r · 2024-12-21

**Metareview:**

(a) This paper introduces HAMSTER, a hierarchical vision-language-action (VLA) model for robot manipulation that aims to improve generalization by separating high-level planning from low-level control. The core claim is that hierarchical VLAs are more effective at transferring behavior across domains than standard monolithic VLA models. HAMSTER fine-tunes a VLM to predict 2D paths representing desired robot motions, and trains a path-guided low-level policy. Experiments in simulation and real-world settings (tabletop manipulation tasks) demonstrate that HAMSTER outperforms monolithic VLA models and baseline imitation learning methods in terms of generalization to unseen scenarios, visual variations, language variations, and spatial variations.

(b) Strengths: The paper provides a thorough description of the experiments, including the different datasets, tasks, evaluation metrics, and baseline methods, making the results easily interpretable and reproducible. The modular design of HAMSTER allows flexibility in choosing different low-level control policies, making it adaptable to various robotic systems and tasks.

(c) Weaknesses:
- The paper emphasizes a lot on the proposal of the hierarchical policy structure. However, hierarchical VLA structure itself is not really new. RT-Trajectory, Code-as-Policies, SayCan, as cited in the paper, can all be considered as such instances, though they might not be trained at both high-level and low-level. I'd think the focus should be on claiming using 2D trajectory/path as an interface to communicate the high level and the low level. I suspect the paper is over claiming its scope.
- The also paper doesn’t seem to address some core challenges in using VLMs for complex manipulation.

(d) The decision is to accept. While I'm a bit concerned about the scope of the paper not being described properly , the averaged review rating leans to accepting the paper.

**Additional Comments On Reviewer Discussion:**

During the rebuttal period, the reviewers raised the following points:
- A reviewer raised concerns about HAMSTER's ability to perform 3D reasoning given its reliance on 2D path representations. In response, the authors argued that while their VLM uses monocular RGB input, it can still address some 3D reasoning challenges, such as inferring the location of a drawer based on instructions like "top," "middle," or "bottom"; They further provided experiment videos demonstrating HAMSTER's ability to handle tasks requiring 3D reasoning.
- Several reviewers questioned the novelty of the work and asked for comparisons with existing methods like RT-Trajectory. The authors conducted a human evaluation comparing HAMSTER with RT-Trajectory, showing that HAMSTER's fine-tuned VLM produces better-ranked path predictions.
- Some reviewers asked for further analysis of HAMSTER's design choices and failure analysis. The authors added correspondingly.

The reviewers' feedback led to improvements in the paper. The authors addressed most concerns.

---

### Decision · Program_Chairs · 2025-01-22

Accept (Poster)